# Probing Visual Concepts in Lightweight Vision-Language Models for Automated Driving

**Nikos Theodoridis**                                      *theodoridis.nikolaos@ul.ie*
*Department of Electronic and Computer Engineering*
*University of Limerick*

**Reenu Mohandas**                                           *reenu.mohandas@ul.ie*
*Department of Electronic and Computer Engineering*
*University of Limerick*

**Ganesh Sistu**                                          *ganesh.sistu@valeo.com*
*Valeo Vision Systems*

**Anthony Scanlan**                                         *tony.scanlan@ul.ie*
*Department of Electronic and Computer Engineering*
*University of Limerick*

**Ciarán Eising**                                          *ciaran.eising@ul.ie*
*Department of Electronic and Computer Engineering*
*University of Limerick*

**Tim Brophy**                                             *tim.a.brophy@ul.ie*
*Department of Electronic and Computer Engineering*
*University of Limerick*

**Reviewed on OpenReview:** *https://openreview.net/forum?id=HlBBy19ojC*

## Abstract

The use of Vision-Language Models (VLMs) in automated driving applications is becoming increasingly common, with the aim of leveraging their reasoning and generalisation capabilities to handle long-tail scenarios. However, these models often fail on simple visual questions that are highly relevant to automated driving, and the reasons behind these failures remain poorly understood. In this work, we examine the intermediate activations of VLMs and assess the extent to which specific visual concepts are linearly encoded, with the goal of identifying bottlenecks in the flow of visual information. Specifically, we create counterfactual image sets that differ only in a targeted visual concept and then train linear probes to distinguish between them using the activations of five state-of-the-art (SOTA) VLMs, including two training variants for one model. Our results show that concepts such as the *presence* of an object or agent in a scene are explicitly and linearly encoded, whereas other spatial visual concepts, such as the orientation of an object or agent, are only implicitly encoded by the spatial structure retained by the vision encoder, or are not linearly encoded at all. In parallel, we observe that in certain cases, even when a concept is linearly encoded in the model's activations, the model still fails to answer correctly. This leads us to identify two failure modes. The first is *perceptual failure*, where the visual information required to answer a question is not linearly encoded in the model's activations. The second is *cognitive failure*, where the visual information is present but the model fails to align it correctly with language semantics. Finally, we show that increasing the distance of the object in question

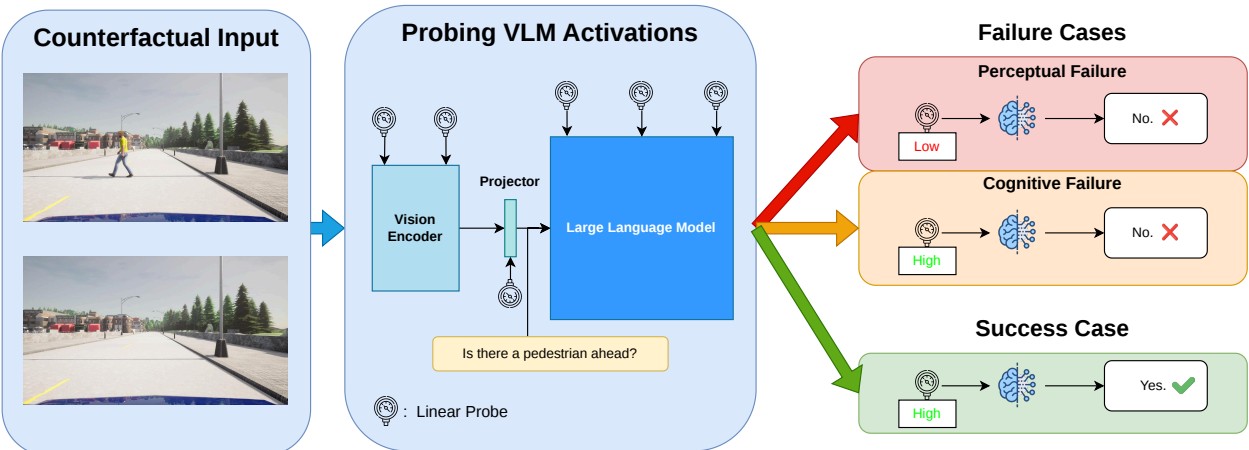

Figure 1: **Framework for tracking visual concept representations across VLM architectures**. Left: The model receives counterfactual input pairs that differ only by a specific visual concept (e.g., the presence of a pedestrian). Middle: Linear probes are trained on intermediate activations to detect if the concept is linearly encoded within the Vision Encoder, Projector, or LLM. Right: Comparing probe accuracy to model output reveals two distinct failure modes: *Perceptual Failure*, where visual information is not linearly encoded in the activations, and *Cognitive Failure*, where the information is encoded (high probe accuracy) but the model fails to align it with language semantics, resulting in an incorrect answer.

quickly degrades the linear separability of the corresponding visual concept. Overall, our findings improve our understanding of failure cases in VLMs on simple visual tasks that are highly relevant to automated driving.[1]

# 1  Introduction

VLMs have shown remarkable progress in the last few years, combining vision encoders with Large Language Models (LLMs) to achieve a multimodal understanding that is applicable and successful in many tasks, from general Visual Question Answering (VQA) (Antol et al., 2015; Goyal et al., 2017; Hudson & Manning, 2019), to expert knowledge (Liu et al., 2024a; Yue et al., 2024; Lu et al., 2022), to document understanding (Mathew et al., 2021; Singh et al., 2019; Liu et al., 2024b), and video understanding (Fu et al., 2025a; Li et al., 2024). The generalisability of these models, together with their strong reasoning capabilities, makes them strong candidates for automated driving applications, where it is often necessary to handle unexpected long-tail scenarios. Prior work has explored different ways of integrating VLMs into automated driving systems, ranging from their use as auxiliary components alongside traditional pipelines (Tian et al., 2024; Jiang et al., 2024a) to stand-alone end-to-end autonomous driving systems (Tian et al., 2024; Xu et al., 2024; Sima et al., 2024; Li et al., 2025; Renz et al., 2025; Hwang et al., 2025; Rowe et al., 2025; Ma et al., 2025; Arai et al., 2025). However, some studies have highlighted the perception limitations of these models, especially for fine-grained and spatial tasks, both in general (Rahmanzadehgervi et al., 2024; Tong et al., 2024; Gou et al., 2024; Kamoi et al., 2025; Kaduri et al., 2025; Yang et al., 2025a) and in traffic scene settings (Theodoridis et al., 2025b). To improve our understanding of these limitations, we propose to examine what happens within the model, and specifically to investigate the role of each model component, namely the vision encoder, the projector, and the LLM (Figure 1, middle), in processing visual information.

Most open-source SOTA VLMs follow a common high-level architecture in which a vision encoder extracts visual features from the image, a projector translates these features to the input space of the LLM, and the LLM processes the complete sequence of visual and textual information to produce an answer or continuation

---

[1]Code available at `https://github.com/D2ICE-Automotive-Research/probing_visual_concepts_in_traffic_scenes`

(Liu et al., 2023; Alayrac et al., 2022; Chen et al., 2024; Bai et al., 2023; Lu et al., 2024). This multi-component architecture makes it impossible to know in advance what went wrong when a VLM fails to answer a simple visual question, since the space of possibilities is very large. For example, a failure could occur because the visual information is not encoded (or is poorly encoded) by the vision encoder, or because the projector degrades the relevant information, or because the information reaches the last layer but the LLM fails to make use of it, or due to a combination of these factors. These possibilities can lead to the same outcome (i.e., the model failing to answer a simple question), but they have distinct causes and should therefore be addressed differently. For this reason, understanding the different failure mechanisms is important for proposing meaningful improvements to VLMs.

There is a growing body of literature that applies interpretability tools to VLMs to understand different aspects of their perception and spatial understanding limitations (Liu et al., 2025; Zhang et al., 2025a;b; Pantazopoulos et al., 2024; Golovanevsky et al., 2025; Neo et al., 2025; Nikankin et al., 2025; Chen et al., 2025). In this study, we use linear probes (Alain & Bengio, 2016; Belinkov, 2022), one of the simplest interpretability methods, to study how well specific visual concepts, that are highly relevant to automated driving, are encoded throughout the architecture of small VLMs. Our goal is to identify where bottlenecks in visual information occur, an approach that is absent from the current literature to the best of our knowledge. Our overall methodology is summarised in Figure 1. We focus on small VLMs due to the computational limitations of the hardware used on vehicles (e.g., NVIDIA Jetson Orin (NVIDIA, 2021)), which does not allow inference with larger models.

The visual concepts we focus on are: a) **presence**, b) **count**, c) **spatial relationship**, and d) **orientation**. More specifically, we create counter-sets of images, using CARLA (Dosovitskiy et al., 2017), that are identical in everything apart from the visual concept in question. We then extract intermediate activations from the VLM (including all vision encoder, projector, and LLM layers) and train simple linear probes to distinguish the different classes of the counter-sets. This allows us to evaluate how well a concept is linearly encoded at each layer and to identify which component is responsible for a failure.

We apply this method to six[2] SOTA small VLMs under 4 billion parameters, which typically correspond to the smallest variants within their respective model families (e.g., Qwen-VL or InternVL), and observe several repeatable patterns. In particular, the presence of an object or agent in the scene is consistently well encoded from the middle layers of the vision encoder through to the final LLM layer, whereas fine-grained spatial concepts, like the orientation of an object, are not explicitly encoded, that meaning there is not a direction for this concept in the model's activation space, throughout the entire model across all six models. However, in some cases and especially for short distances, spatial concepts are implicitly represented through the spatial structure retained in the vision encoder's representations.

We also identify two main modes of failure, which we refer to as **perceptual failure** and **cognitive failure**. In perceptual failure, the visual information necessary for answering a question does not reach the last layer of the model due to a bottleneck earlier in the architecture[3]. In contrast, a cognitive failure occurs when the visual information is well encoded, but the model still fails to answer correctly. These two failure mechanisms are not expected to have distinct impacts on automated driving applications, since both result in similar behaviour, namely the failure to complete a task. However, they arise from different underlying causes and may therefore require different mitigation strategies. For this reason, we argue that distinguishing between these two failure modes is important. Finally, we show that increasing object or agent distance quickly degrades the linear separability of visual concepts in the activation spaces of the models, which is a concerning result for the application of VLMs in traffic scenes, where critical objects and agents are often located at a distance.

Overall, our main contributions are as follows:

1. We create visually perfect counterfactual samples that isolate and target specific visual concepts, namely *presence*, *count*, *spatial relationship*, and *orientation*, which are highly relevant to automated driving.

---

[2]We actually use five distinct models with two training variants for one of them.

[3]Throughout this paper, when we say that visual information is absent or not encoded at a particular part of the model, we mean that it is not linearly encoded, not that it is completely absent.

2. We analyse the flow of visual information throughout the whole architecture of small SOTA VLMs for the targeted concepts, and we identify common patterns of visual bottlenecks across different models.

3. We identify two failure modes that prevent modern VLMs from answering simple visual questions: **perceptual failure** and **cognitive failure**.

4. We show how increasing distance degrades the representation quality throughout the model, even for very simple concepts, like the presence of an object in the scene.

## 2 Related Work

### 2.1 Linear Probes

Linear probes were first introduced by Alain & Bengio (2016) as a means to understand how linear separability of the original classes evolves across the layers of CNNs. Since then, they have evolved into a comparison tool for vision encoders (Chen et al., 2020; Radford et al., 2021; Oquab et al., 2024). In this setting, researchers extract visual features from a vision encoder using ImageNet (Deng et al., 2009) images and then train linear probes on these features, with higher probe accuracy indicating higher-quality visual representations. Closer to our use case, linear probes have also been used for interpretability studies. Nanda et al. (2023) used linear probes to find the board state encoded in the intermediate activations of a transformer-based Othello-playing model. Ravfogel et al. (2020) used linear probes to detect and remove certain biases encoded in word embeddings. Marks & Tegmark (2024) used linear probes, among other techniques, to show that the concept of falsehood is linearly encoded in the internal activations of LLMs. Similarly, Azaria & Mitchell (2023) used non-linear probes to detect the truthfulness of statements in the internal activations of LLMs. Regarding applications in VLMs, Pantazopoulos et al. (2024) used linear probes to compare two different projectors, both based on the Q-Former architecture (Li et al., 2023a), in terms of their ability to preserve spatial information, and found that neither projector preserves fine-grained spatial information. We extend the application of linear probes by applying them throughout the entire VLM architecture, rather than only to specific components.

### 2.2 VLMs in Automotive

A growing body of literature explores the use of VLMs for automated driving tasks. Broadly, there are two main approaches to integrating VLMs into automated driving systems. On one hand, some works employ VLMs as part of a dual-system architecture (Tian et al., 2024; Jiang et al., 2024a), where the VLM is responsible for making high-level decisions about the vehicle's behaviour, which are then consumed by a specialised autonomous driving module to predict the future trajectory. On the other hand, other works aim to use VLMs for end-to-end autonomous driving without relying on additional components (Tian et al., 2024; Xu et al., 2024; Sima et al., 2024; Li et al., 2025; Renz et al., 2025; Hwang et al., 2025; Rowe et al., 2025; Ma et al., 2025; Arai et al., 2025)[4]. Within this second approach, two subcategories can be identified. In the first, driving is formulated as a problem in the language space, where the model predicts the future trajectory and or vehicle control signals as text tokens (Tian et al., 2024; Xu et al., 2024; Sima et al., 2024; Hwang et al., 2025; Rowe et al., 2025; Ma et al., 2025). In the second, driving-specific decoders are introduced, enabling predictions directly in a continuous numerical space (Li et al., 2025; Arai et al., 2025; Renz et al., 2025). Despite the substantial body of work applying these models to automated driving pipelines, their perception limitations and the underlying causes of these limitations remain insufficiently documented.

### 2.3 Perception limitations in VLMs

A growing body of literature studies the perception limitations of VLMs. Rahmanzadehgervi et al. (2024) examined the performance of several SOTA VLMs on simple visual tasks that humans can solve almost instantaneously, such as identifying a circled letter within a word or counting overlapping circles. Their

---

[4]Tian et al. (2024) try both approaches.

findings indicate that even advanced models, including GPT-4o (OpenAI, 2024) and Claude-3.5-Sonnet (Anthropic, 2024), perform poorly on these seemingly trivial questions. Similarly, Kamoi et al. (2025) proposed VisOnlyQA, a benchmark specifically designed to evaluate the pure visual perception capabilities of VLMs independently of reasoning. Evaluations on GPT-4o, Gemini 1.5 Pro (Sundar Pichai, 2024), and InternVL2-76B (Team, 2024) revealed consistently weak perceptual performance across all models. Notably, the authors also showed that fine-tuning on perception-focused benchmarks alone was insufficient to meaningfully mitigate these deficiencies. Yang et al. (2025a) introduced VSI-Bench, a benchmark for evaluating the spatial understanding and reasoning of VLMs, and found that, while not trivial, the spatial intelligence of current VLMs falls far behind that of humans.

## 2.4  Interpretability of VLMs

Another line of research attempts to interpret the perception capabilities of VLMs. Kaduri et al. (2025) studied the attention patterns of VLMs and found that the last token in the sequence mainly attends to text tokens. This suggests that visual information does not flow directly from visual tokens to the last token, but instead passes through text tokens as an intermediate step. Neo et al. (2025) intervened in intermediate activations to examine whether object information is encoded locally or globally, and used the logit lens to analyse how well visual tokens encode the object of the corresponding patch throughout the model's layers, focusing only on the LLM component. Golovanevsky et al. (2025) introduced counter-pairs of images and applied activation patching to study the different roles of cross-attention and self-attention in BLIP (Li et al., 2022) and LLaVA (Liu et al., 2023), respectively. Nikankin et al. (2025) studied the gap between how text and visual tasks are processed by the model and found that visual representations only become meaningful, that is, well aligned with language, deep in the network. As a result, there is limited capacity for further processing. Focusing more specifically on spatial understanding and reasoning, Chen et al. (2025) investigated why spatial reasoning is difficult for VLMs by analysing their attention patterns. They found that models often fail to align attention with relevant objects in the image and instead rely heavily on language priors.

A slightly different line of work studies the discrepancy between the visual information that appears to be available to VLMs and whether they actually use it. Liu et al. (2025), and Zhang et al. (2025a) showed that VLMs often attend to the correct regions of the image even when producing incorrect answers. This suggests that they may have access to the necessary visual information but still fail to use it correctly. Vompa et al. (2025) showed that, in most cases, the generative performance of VLMs is below the linear separability of their own activations by comparing the generated answers to the linear separability of activations from the final vision encoder and LLM layers. Rajaram et al. (2025) used linear probes and sparse autoencoders (SAEs) (Bricken et al., 2023) to study whether representations are linearly structured within LLM layers and how they evolve. Closer to our work, Zhang et al. (2024) showed that VLMs underperform in image classification tasks compared to their vision encoders. They trained linear probes on the activations of the last LLM layer and showed that these perform similarly to the vision encoder, indicating that the relevant information is present throughout the LLM but is not effectively utilised by the model. Similarly, Fu et al. (2025b) show that VLMs perform worse than their vision encoders on several vision-centric tasks, and that this gap is not due to degradation of visual information in the projector or LLM. Instead, they show that visual information is preserved and accessible across these components, highlighting that VLMs can fail even when the relevant visual information is encoded. This aligns to some degree with what we describe as cognitive failure in our work.

Our work differs from the studies above in the following ways: 1) we study the layer-wise evolution of linearly encoded visual concepts across the entire VLM architecture, including the vision encoder; 2) we carefully isolate the visual concepts of interest by constructing identical counter-sets of images that differ only in the corresponding visual concept, a setting that is largely absent from the current literature to the best of our knowledge; 3) we focus on specific visual concepts that are highly relevant for understanding traffic scenes; and 4) we explicitly study small, lightweight VLMs, as these are suitable for deployment on on-vehicle hardware.

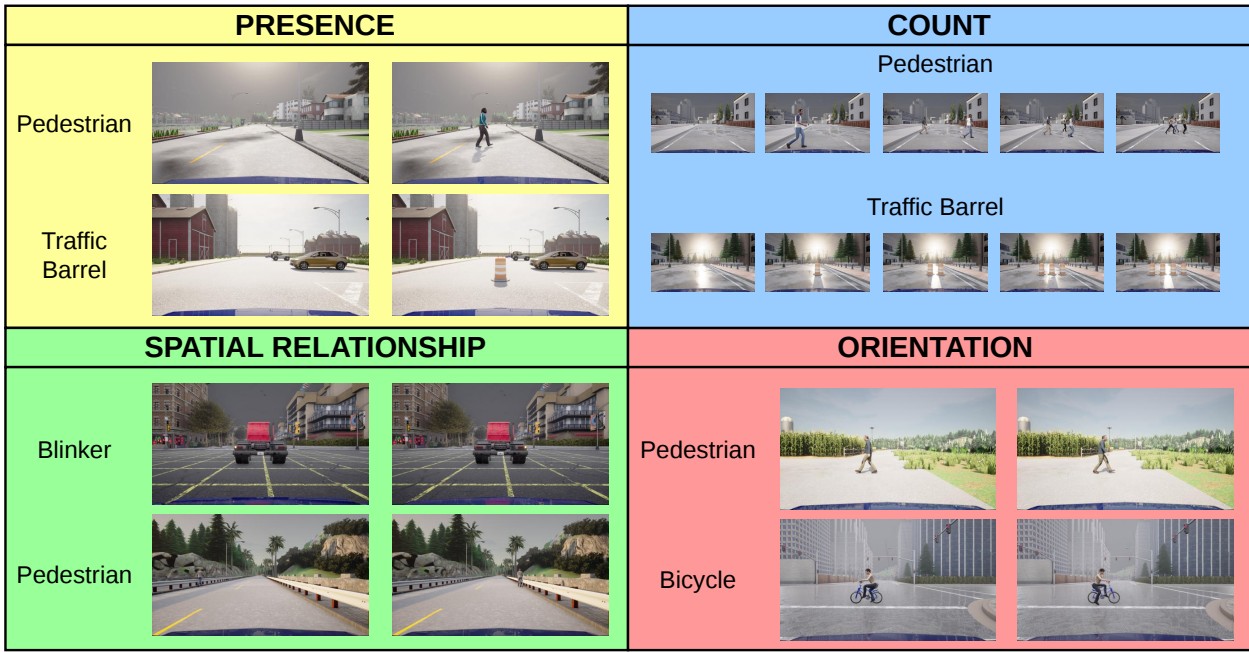

Figure 2: **Counterfactual sets of images representing four basic visual concepts**: *Presence, Count, Spatial Relatioinship, Orientation*

## 3 Counterfactual Sets

Similar to works that study linear concept directions in the embedding or activation space of LLMs (Park et al., 2024; Burns et al., 2023; Jiang et al., 2024b; Marks & Tegmark, 2024), we need counterfactual sets of inputs to determine whether a targeted visual concept is encoded in a model's activations. Since we are interested in mapping the visual information flow, we require the difference between the counterfactual pairs to lie in the visual part of the input. Specifically, we aim for counterfactual images that are identical in every respect except for the visual concept under examination. This allows us to isolate the target concept as effectively as possible and mirrors the use of textual counterfactual inputs in LLM interpretability research, a methodology that has not yet been translated to the same level of granularity in the visual domain.

More formally, given a concept $c$ with states $c_1...c_n$, we need a set of images $I = I_{c_1},...I_{c_n}$ in which all images are identical in everything they depict except for the state of concept $c$. Because there is currently no dataset that offers such counterfactual images for traffic scenes, we use CARLA Dosovitskiy et al. (2017) to create these images. In §3.1, we present the visual concepts we focus on and the counterfactual sets of images generated for each of them, and in §3.2, we discuss details of the data generation process that ensure high quality and alignment with our research objectives.

### 3.1 Visual Concepts and Data

We focus our study on four basic visual concepts that are highly important for correctly perceiving a traffic scene. These are **presence**, *i.e., whether something is present in the scene*, **count**, *i.e., how many instances of an object are present in the scene*, **spatial relationship**, *i.e., where something is located in the scene in relation to something else*, and **orientation**, *i.e., the orientation of an object in the scene* (left- or right-facing). A typical traffic scene contains multiple objects or agents, which relate to *presence* and *count*, arranged within a three-dimensional environment, which relates to *spatial relationships*, and exhibiting various orientations. We therefore argue that successfully encoding these visual concepts is necessary, although not sufficient, for a proper understanding of a traffic scene. For each of these concepts, we created counterfactual image sets that differ only with respect to the specific visual concept under consideration.

Table 1: **VLMs used in the experiments**. FT indicates whether the model is a general-purpose vision-language model (General) or has been additionally fine-tuned for a specific task/domain, such as spatial reasoning (Spatial) or autonomous driving (AD).

| VLM | Param (B) | Vision Encoder | Projector | Language Model | FT |
|---|---|---|---|---|---|
| Ovis2.5-2B | 2.57 | SigLIP2-400M | MLP[5] (436M) | Qwen3-1.7B | General |
| InternVL3.5-2B | 2.35 | InternViT-300M-v2.5 | MLP (12.6M) | Qwen3-1.7B[6] | General |
| Qwen3-VL-2B | 2.13 | SigLIP2-Large (300M) | MLP (100M) | Qwen3-1.7B | General |
| VST-3B | 3.75 | QwenViT (632M) | MLP (36.7M) | Qwen2.5-3B | Spatial |
| DriveFusionQA | 3.75 | QwenViT (632M) | MLP (36.7M) | Qwen2.5-3B | AD |

To materialise these theoretical concepts, we created two data categories for each, which are depicted in Figure 2. More specifically, for *presence* we have **Presence-1** and **Presence-2**, which depict the presence or absence of a pedestrian and a traffic barrel in the scene, respectively. For *count*, we have **Count-1** and **Count-2**, which depict multiple instances of either pedestrians or traffic barrels in the scene. The count takes values from 0 to 4. For *spatial relationship*, we have **Spatial-1**, which depicts a truck ahead with one of its blinkers active, and **Spatial-2**, which depicts a pedestrian walking along the road either on the left or on the right side. Finally, for *orientation understanding* we have **Orientation-1** and **Orientation-2**, which depict a pedestrian and a bicycle, respectively, moving left or right from the camera's point of view.

### 3.2 Generation Details

For each data category, we created multiple versions of the same sample in which the object or agent of interest appears at varying distances from the camera, specifically at 5, 10, 20, 30, 40, and 50 meters. The data belonging to Spatial-2 do not include versions at 5 meters, as the pedestrian was falling out of the field of view of the camera at such close proximity. We used multiple maps in CARLA (`Town01`–`Town07`, `Town10HD`, `Town12`, and `Town15`) and multiple weather conditions, while ensuring that the weather did not make it too difficult to see the object or agent of interest. In particular, we avoided night-time conditions for all categories except Spatial-1, which can benefit from night-time conditions, as it depicts an illuminated blinker. For Spatial-1, we avoided sunset conditions, as we observed that the blinker light can become very difficult to see in such cases, especially at long distances. Exceptionally, for Spatial-1, we didn't use `Town12`, while for Spatial-2, we used only `Town01`, `Town02`, and `Town07`, because the remaining maps depict very wide roads, which made it difficult to successfully depict the desired concept. In total, we created 500 samples per class per distance for each data category. All images were captured using CARLA's standard RGB camera with default settings, including a field of view of 90 degrees, and were saved at a resolution of $1920 \times 1080$.

## 4 Methodology

In this section, we describe the experiments we conducted. In §4.1 we present the VLMs selected for our experiments, in §4.2 we briefly explain how we evaluate the selected models on the counter-sets of images, in §4.3 we describe how we extracted the intermediate activations during the forward pass of these models, and finally in §4.4 we describe in detail how we trained the linear probes on the extracted activations.

### 4.1 Models

We used five under-4-billion-parameter models for our experiments, which are presented in Table 1. All of these models follow the same high-level architecture of vision encoder $\rightarrow$ projector $\rightarrow$ LLM (Figure 1, middle).

---

[5]The Ovis2.5 technical report states that visual features are projected into the LLM input space via a weighted sum over a learned visual embedding table. Mathematically, however, this mechanism is equivalent to a softmax-activated MLP projector with a significantly expanded hidden dimension.

[6]For InternVL3.5, they unbind the embedding and unembedding matrices of Qwen3, resulting actually in a model with 2B parameters instead of 1.7B.

In this setup, the visual input first passes through a vision encoder, usually a Vision Transformer (Dosovitskiy et al., 2021), to produce visual features. These features are then mapped by a projector, typically an MLP, into the input space of the LLM, producing visual embeddings[7]. Finally, the visual embeddings and the text embeddings are processed together as a single sequence by the LLM.

We selected Ovis2.5 (Lu et al., 2025), InternVL3.5 (Wang et al., 2025), and Qwen3-VL (Bai et al., 2025) because they are all well-known SOTA models and exhibit key architectural differences, as discussed below, which enable direct comparisons of how these differences influence the linear encoding of the studied visual concepts. We additionally included VST (Yang et al., 2025b) because it was trained specifically to improve spatial understanding and reasoning, and DriveFusionQA (Samir & Team, 2026) because it was fine-tuned specifically for autonomous driving scenarios. These two models allow us to assess whether the patterns observed in general-purpose models also emerge in domain-specialised models, or whether training on task-relevant data is sufficient to mitigate the limitations of general-purpose models. Despite their similarities, there are the following key differences between the models:

1. InternVL3.5 uses learned positional embeddings, which restrict the size of the images it can take as input. To work around this limitation, when higher-resolution images are used, the model splits them into an adequate number of tiles, each equal to the size that InternVL3.5 can process, and these tiles are processed in parallel. In contrast, Ovis2.5, Qwen3-VL, VST, and DriveFusionQA make use of Rotary Position Embeddings (RoPE), which allow them to natively process images of any size up to a limit. For images that exceed this limit, the models downsize the image while retaining the aspect ratio.

2. Ovis2.5 uses significantly more parameters for the projector component compared to the other models. This is because it follows a different design choice. Instead of using a simple MLP as a projector, it first constructs a learnable visual embedding table, analogous to the text embedding table. The visual features from the final layer of the visual encoder are then projected into a probability distribution over this "visual vocabulary", and a weighted sum of the visual embeddings is computed to produce the final visual representations that are passed to the LLM.

3. Qwen3-VL uses multiple projectors, which explains the 100M projector parameters reported in Table 1, to inject visual features at different levels of granularity into the LLM. Specifically, in the 2B variant used in our experiments, visual features from layers 6, 12, 18, and 24 (the final layer) of the vision encoder are projected into the LLM input space, rather than relying solely on features from the final layer, which is the conventional approach. Incorporating early- and intermediate-layer features may help preserve fine-grained visual information that could otherwise be lost in deeper representations.

It is also important to note that we used two different versions of the VST model, namely VST-SFT and VST-RL. The first is the model that only went through the supervised fine-tuning stage, while the second also went through a reinforcement learning training stage. We used both versions in our experiments in order to see if reinforcement learning changes the way these models encode visual information when answering very simple visual questions.

## 4.2 Model evaluation

The first step is to evaluate the models on the data in order to later be able to compare their performance to that of the linear probes. We evaluate the models only on the test data (see §4.4), namely `Town07` for Spatial-2 and `Town15` for all other data categories. We provide the image and the corresponding question to the model as input, perform a single forward pass, and select as the predicted answer the token with the highest probability, which we then compare to the ground truth answer. We used the following prompt structure:

---

[7]This terminology and distinction between visual features and visual embeddings is not standard in the literature; however, we find it useful for distinguishing between visual representations before and after the projector.

Table 2: **Questions used for each data category**.

| Category | Question | Answers |
|---|---|---|
| Presence-1 | Is there a pedestrian ahead? | Yes \| No |
| Presence-2 | Is there a traffic barrel ahead? | Yes \| No |
| Count-1 | How many pedestrians are ahead? | Zero \| One \| Two \| Three \| Four |
| Count-2 | How many traffic barrels are ahead? | Zero \| One \| Two \| Three \| Four |
| Spatial-1 | Which of the truck's blinkers is on? | Left \| Right |
| Spatial-2 | On which side of the road is the pedestrian walking? | Left \| Right |
| Orientation-1 | In which direction is the pedestrian walking? | Left \| Right |
| Orientation-2 | In which direction is the bicycle moving? | Left \| Right |

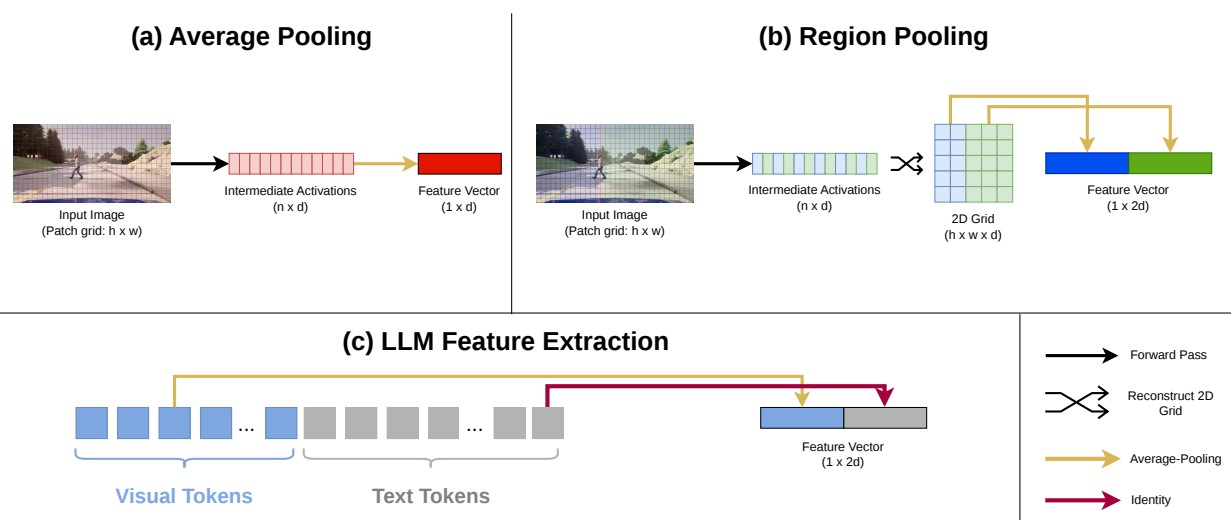

Figure 3: **Activation extraction methodology**. a) We apply average pooling to all patch vectors to obtain a single vector representation of the intermediate activations. b) We split the image at a selected point and apply average pooling to the left and right regions independently. We then concatenate the resulting vectors to form a single representation of the intermediate activations while retaining minimal spatial structure. c) We apply average pooling to the visual token vectors within the LLM and concatenate the result with the activation of the last token.

"Strictly answer with a single word only: `<question>` Possible answers: [`<answers>`]"

The questions and possible answers for each data category are included in Table 2. We also evaluated the models with constrained decoding over the answers set (more details in Appendix A).

## 4.3 Activation extraction

In order to train linear probes on the intermediate activations of the models, we first need to extract and save those activations. More specifically, we extracted the activations from every transformer block in the vision encoder and the LLM, as well as the output of the projector module. All activations were saved as PyTorch tensors of type `float32`.

The intermediate activations of VLMs are two-dimensional, with shape $seq\_len \times hidden\_dim$[8]. Therefore, we cannot directly train linear probes on them, as this requires a one-dimensional representation. A naive approach would be to simply flatten the activations. However, storing flattened activations for thousands

---

[8]Ignoring the batch size.

of images across dozens of layers and six models would require an extremely large amount of storage. Consequently, the only feasible solution is to compress the activations. The specific compression strategy is a design choice that directly influences the claims we can make based on our results. By compressing the activations, we inevitably discard some information and thus lose a complete view of what is encoded in the model's activations. Below, we describe which compression strategies we chose for each model component and clarify the types of claims that can be supported based on them.

### 4.3.1 Vision Encoder

Let $\mathbf{V}^{(l)} \in \mathbb{R}^{n \times d_V}$ denote the output of layer $l$ of the vision encoder, where $n$ is the number of patches ($\mathbf{V}^{(l)} \in \mathbb{R}^{(n+1) \times d_V}$ if a [CLS] token is used by the model) and $d_V$ is the hidden dimension of the vision encoder. A common strategy to reduce the two-dimensional representation into a one-dimensional representation, that is followed in other works that focus on isolated vision encoders (Oquab et al., 2024), is to either use only the activation of the [CLS] token, $\mathbf{v}_{cls}^{(l)} = \mathbf{V}_0^{(l)}$, as this is designed to be representative of the whole image and thus the entire sequence of activations, or the concatenation of $\mathbf{v}_{cls}^{(l)}$ with an element-wise average across all patch activation vectors. However, in our case, this approach is not possible, as only InternVL3.5 utilises a [CLS] token. Furthermore, even when present, the [CLS] token is discarded before the projector and is not used by the LLM later. Consequently, our first approach is to use only the element-wise average of the patch embeddings as the representation for layer $l$. Formally, if we denote the patch vectors as $\mathbf{v}_i^{(l)}$ (the rows of $\mathbf{V}^{(l)}$), the extracted activation vector is

$$\mathbf{v}^{(l)} = \frac{1}{n} \sum_{i=0}^{n-1} \mathbf{v}_i^{(l)} \in \mathbb{R}^{d_V}. \tag{1}$$

If a [CLS] token is used, then the summation runs from 1 to $n$ instead. This approach is depicted in Figure 3(a).

Because this averaging operation removes all spatial structure, the resulting representation can only reveal whether a visual concept is explicitly and linearly encoded in individual patch embeddings. For example, if the linear probe achieves low accuracy on Orientation-1, we can conclude that the pedestrian's orientation is not explicitly and linearly encoded in individual patch activations. However, we cannot conclude that the concept is absent from the full set of activations. It may still be implicitly encoded in the spatial configuration of multiple patches, which could, in principle, allow orientation to be inferred. Therefore, this approach limits the strength of the claims that can be made for concepts that require spatial understanding.

To address this limitation, we introduce a second activation extraction strategy for tasks that require spatial understanding, namely *spatial relationship* and *orientation* tasks. Since these tasks depend on left–right distinctions, we partition the image into left and right regions, adequate for distinguishing the counterexamples (Figure 3(b)). At an intermediate layer, we reconstruct the 2D grid corresponding to the spatial structure of the input image[9] and compute the average of the activation vectors in the left region to obtain $\mathbf{v}_{left}^{(l)}$. Similarly, we average the activations in the right region to obtain $\mathbf{v}_{right}^{(l)}$. The final representation for layer $l$ is the concatenation of these two vectors:

$$\mathbf{v}_{sp}^{(l)} = [\mathbf{v}_{left}^{(l)}, \mathbf{v}_{right}^{(l)}] \in \mathbb{R}^{2d_V} \tag{2}$$

This strategy preserves minimal spatial structure sufficient to distinguish spatial counterexamples. Under this representation, if a linear probe achieves low accuracy, we can more confidently conclude that the visual concept is not linearly encoded in the model's activations at that layer. We refer to this extraction strategy as region-pooling to distinguish it from the average-pooling described previously.

---

[9]For InternVL3.5, we reconstruct the 2D grid using the activation vectors from the first eight tiles only, discarding the thumbnail tile.

### 4.3.2 Projector

Similarly, given the output of the projector $\mathbf{P} \in \mathbb{R}^{n' \times d_L}$, where $n'$ is the number of visual embeddings (usually $n' = n/4$) and $d_L$ is the hidden dimension of the LLM, we define $\mathbf{p} \in \mathbb{R}^{d_L}$ analogously to (1) and $\mathbf{p}_{sp} \in \mathbb{R}^{2d_L}$ analogously to (2).

### 4.3.3 LLM

Let $\mathbf{L}^{(l)} \in \mathbb{R}^{t \times d_L}$ be the output of layer $l$ of the LLM, where $t$ is the total number of tokens (visual and textual) in the sequence. Given that we are interested in tracking the visual information within the model, one might think that focusing only on the visual tokens is sufficient. However, because of the attention modules, visual information can propagate from the visual tokens to subsequent text tokens, so neglecting the latter would provide only a partial view. Additionally, for the visual information to be useful in answering the question, it must reach the last text token in the sequence, whose activation vector is used to select the next token. With this in mind, we define the one-dimensional representation of layer $l$ of the LLM as $\mathbf{l}^{(l)} = [\mathbf{l}_{visual}^{(l)}, \mathbf{l}_{t-1}^{(l)}] \in \mathbb{R}^{2d_L}$, where

$$\mathbf{l}_{visual}^{(l)} = \frac{1}{|K|} \sum_{i \in K} \mathbf{l}_i^{(l)} \in \mathbb{R}^{d_L}, \tag{3}$$

where $K$ is the set of visual token indices, and $\mathbf{l}_{t-1}^{(l)}$ is simply the last token in the sequence. This representation captures two aspects: first, whether the visual information relevant to the target task is explicitly and linearly encoded in the visual tokens; second, whether it is explicitly and linearly encoded in the activation of the final token. Even if the information is not linearly encoded in the visual tokens, it may still become linearly encoded in the final token through the LLM transforming non-linear visual features into a linearly accessible representation. The above extraction strategy is depicted in Figure 3(c).

As with the vision encoder and projector, we define a second representation for spatial tasks to retain partial spatial information. This representation is $\mathbf{l}_{sp}^{(l)} = [\mathbf{l}_{visual(sp)}^{(l)}, \mathbf{l}_{t-1}^{(l)}] \in \mathbb{R}^{3d_L}$, where $\mathbf{l}_{visual(sp)}^{(l)}$ is simply constructed analogously to (2).

Finally, we extract activations from the post-layernorm layer of the model, following (3) and, for the spatial variation, (2). We do this because activations at this layer allow us to assess whether the model encodes the target concept at all. In earlier layers, a concept may still be encoded non-linearly even if a linear probe fails to detect it. By contrast, at the post-layernorm stage, all information must be linearly encoded, since these activations are passed only through a linear projection, the language head, to be mapped to the vocabulary. Consequently, any information that remains non-linearly encoded at this stage cannot influence the model's output and is therefore not useful.

### 4.4 Training Linear Probes

Once the activation vectors are extracted, we train linear probes for a classification task. Each probe is a linear classifier applied to standardised activations. Specifically, for each model, data category, distance, and layer, we compute the mean and standard deviation activation vectors on the training set and use them to standardise the activations before feeding them to the linear probes during training, validation, and testing. More formally, given an extracted activation vector $\mathbf{f}^{(l)} \in \mathbb{R}^d$ at layer $l$, we first compute

$$\tilde{\mathbf{f}}^{(l)} = \frac{\mathbf{f}^{(l)} - \boldsymbol{\mu}^{(l)}}{\boldsymbol{\sigma}^{(l)}}, \tag{4}$$

where $\boldsymbol{\mu}^{(l)}$ and $\boldsymbol{\sigma}^{(l)}$ are the mean and standard deviation vectors computed on the training set. The probe then computes the logits as

$$\mathbf{z}^{(l)} = W\tilde{\mathbf{f}}^{(l)} + b, \tag{5}$$

where $W \in \mathbb{R}^{c \times d}$ and $b \in \mathbb{R}^c$ are the learned parameters of the linear probe. Here, $d$ is the hidden dimension and $c$ is the number of classes. For binary classification tasks, we use a single output logit, i.e., $c = 1$.

Most previous work (Alain & Bengio, 2016; Nanda et al., 2023; Ravfogel et al., 2020; Pantazopoulos et al., 2024) trains linear probes directly on raw activations, without first standardising them. However, we observed that the per-dimension variances of the extracted activation vectors, computed across samples, can differ substantially across layers. This variation can potentially affect the conditioning of the probe optimisation problem, making comparisons across layers less reliable. We therefore chose to standardise the activations before training the probes. Furthermore, because standardisation is an affine transformation and our linear probes include a bias term, it does not alter whether a concept is linearly separable.

Since we also aim to capture the effect that object distance has on the representation quality of each visual concept, we repeat the experiments for each distance. We use `Town01-Town07` and `Town10HD` for training, that is, 400 samples per class per distance, `Town12` for validation, that is, 50 samples per class per distance, and `Town15` for testing, that is, again, 50 samples per class per distance. For Spatial-1, `Town10HD` was used as the validation set instead. For Spatial-2, we use `Town01` and `Town02` for both training and validation, and `Town07` for testing, while keeping the number of samples the same as above.

Similarly to Oquab et al. (2024), we evaluate multiple learning rates in the range from $1e^{-4}$ to $5e^{-1}$ and only consider the test accuracy corresponding to the best one, that is, the learning rate that achieved the highest validation accuracy. To reduce stochastic fluctuations in accuracy across layers, we repeat this process ten times and report the average test accuracy of the best probe from each run, along with the standard deviation across the ten runs. We use AdamW (Loshchilov & Hutter, 2019) with default parameters to optimise the probes.

## 5 Results and Analysis

In this section, we present the results of the above experiments. In §5.1, we go through the results of the linear probe training for each visual concept and data category. In §5.2, we conduct a series of experiments to validate that the probes indeed learned the direction of the target visual concepts in the activation spaces of the models instead of data-specific statistics, and that the learned direction causally affects the model's output. In §5.3, we first present the accuracy of the models in comparison to the accuracy of the probes and then, based on this comparison, argue that it is meaningful to think in terms of two failure modes: a *perceptual failure* and a *cognitive failure*.

### 5.1 Main Results

We can see the results of the linear probes trained on the average-pooled activations in Figure 4 for all models, data categories, layers, and distances. For completeness, Figure 5 shows the results of the linear probes trained on the region-pooled activations for the spatial and orientation tasks. The colour brightness indicates the average chance-corrected accuracy of all ten probes trained on each layer's activations at a specific distance. We define chance-corrected accuracy $a'$ by scaling the observed accuracy relative to random chance, a formulation mathematically equivalent to Cohen's Kappa (Cohen, 1960), bounded at zero:

$$a' = \begin{cases} \frac{a_o - a_c}{1 - a_c}, & \text{if } a_o > a_c \\ 0, & \text{if } a_o \leq a_c \end{cases} \tag{6}$$

where $a_o$ is the observed accuracy and $a_c$ is the chance accuracy. We report chance-corrected accuracy for a fair comparison between the *count* data, which have five classes and hence a chance accuracy of 20%, and all other data, which have two classes and hence a chance accuracy of 50%. The standard deviation of probe accuracies across the ten runs for a fixed model, data category, layer, and distance is generally low, ranging from 0.00 to 0.09, with an average value of 0.02. Below, we discuss the main conclusions from these results for each visual concept.

An important consideration when interpreting the Qwen3-VL heatmaps is that, unlike those of the other models, they do not represent a strictly sequential flow through the vision encoder, projector, and LLM. As

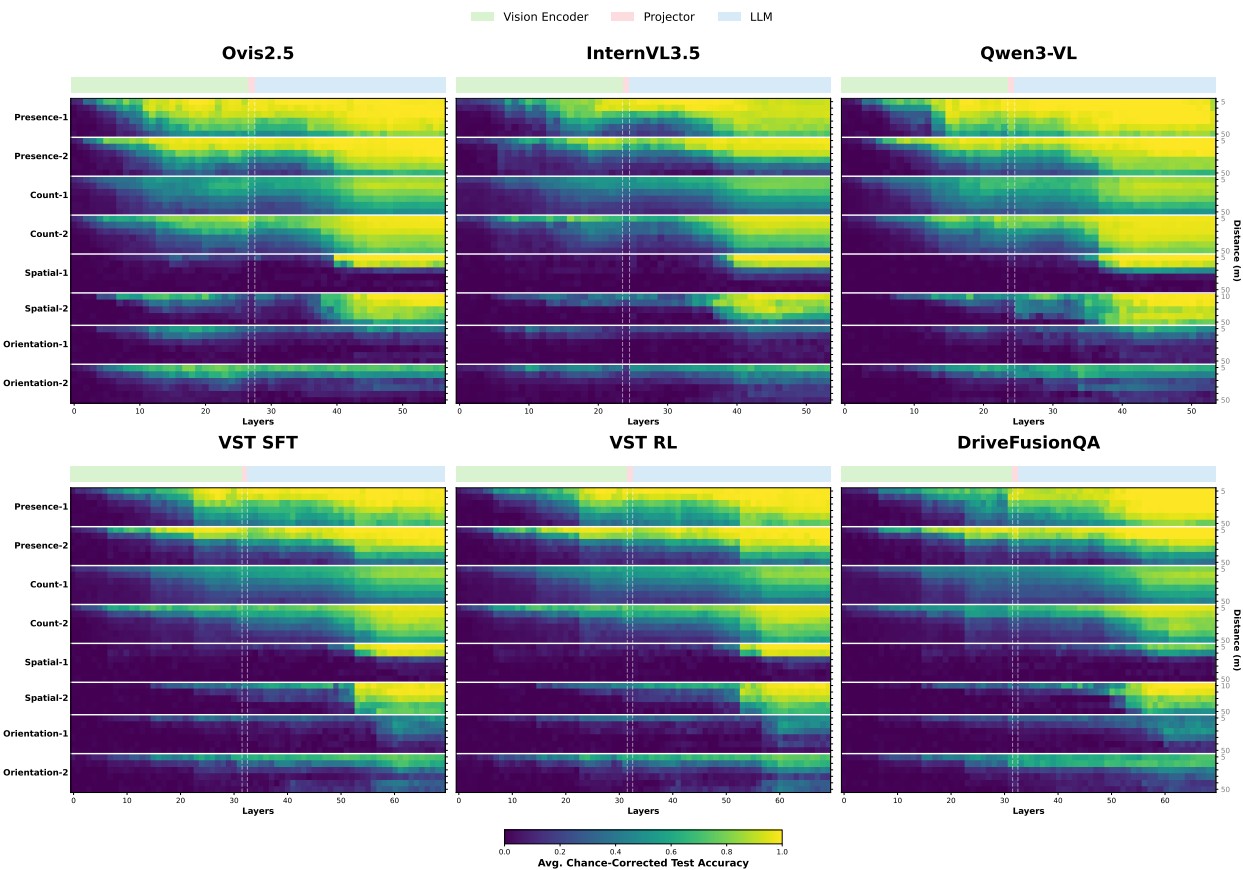

Figure 4: **Linear separability of visual concepts across models and layers based on average-pooled activations**. Linear separability for each model, layer, and distance is measured as the average test-set accuracy across ten linear probes trained under the same setting.

discussed in §4.1, Qwen3-VL uses four projectors to inject features from different vision-encoder depths into selected LLM layers, thereby introducing additional pathways for visual information. For visual consistency with the other models, Figures 4 and 5 show only the output of the projector applied to the final vision-encoder layer; the other three projector outputs cannot be assigned a single sequential position on the heatmap x-axis and are therefore omitted.

### 5.1.1 Presence

As shown in Figure 4, the results are very similar for both Presence-1 and Presence-2 data across all models. More specifically, we observe that in the very first layers, the linear probe's accuracy is at chance level, but then it increases quickly, and in the middle-to-late layers of the vision encoder, the accuracy is already very close to perfect for short distances (5-20 meters). For longer distances (30-50 meters), however, the accuracy remains lower, showing that the farther away from the camera an object is, and hence the smaller it appears in the image, the less linearly its presence is encoded in the activations of the vision encoder.

Within the LLM, the quality of the encoding initially remains at the same level but then increases in the middle layers for longer distances. For example, for Presence-1 at 50 meters, Ovis2.5 exhibits a 32% increase in chance-corrected accuracy between the first and last LLM layers. This suggests that the LLM component, aided by its access to the question and thus knowledge of what to look for, can improve how the presence of an object is represented in the model. However, despite this increase, the representation quality at longer

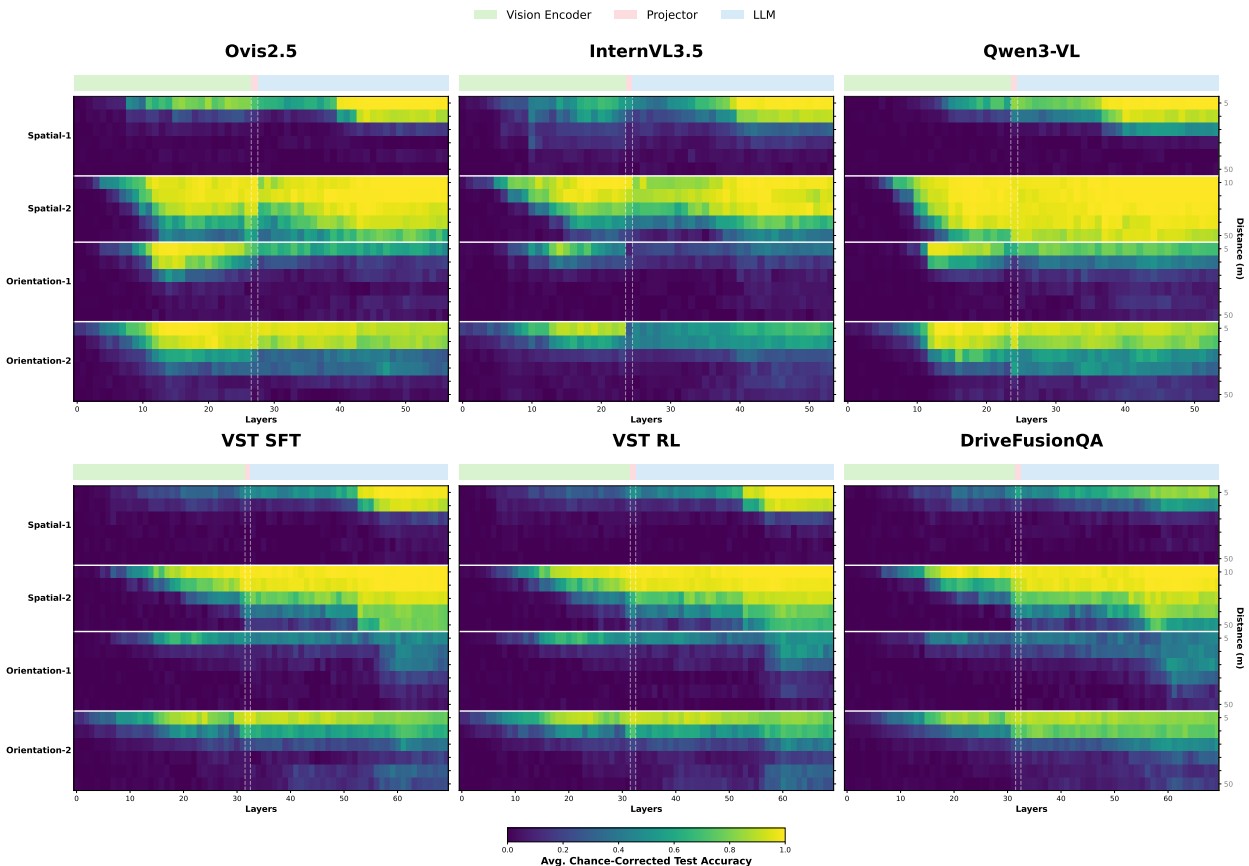

Figure 5: **Linear separability of visual concepts across models and layers based on region-pooled activations**. Linear separability for each model, layer, and distance is measured as the average test-set accuracy across ten linear probes trained under the same setting.

distances still remains lower compared to short distances. Some additional points worth mentioning are the following:

- The representation quality of pedestrian presence in the scene is higher than that of traffic barrel presence, especially at longer distances. A plausible explanation is the smaller physical size of traffic barrels compared to pedestrians. For instance, at 30 meters, the pedestrian bounding box has a size of $49 \times 92$, whereas the traffic barrel bounding box measures $34 \times 43$ in $1920 \times 1080$ images. Another contributing factor may be the distribution of object classes in the VLMs training data: pedestrians are likely far more frequent than traffic barrels, making the models more familiar with this class.

- The representation quality in InternVL3.5 decreases for long distances in the last few layers of the vision encoder before increasing again after the first few layers of the LLM. A plausible explanation for this behaviour may be related to the fact that InternVL3.5 splits images into tiles and processes them in parallel as a batch within the vision encoder. As a result, visual tokens from different tiles first "communicate" in the LLM, and it may therefore take a few layers for them to form a comprehensive representation.

Overall, these results suggest that the presence of an object in the scene is very well encoded at short distances and less so at larger distances, with the gap between the two being created in the vision encoder.

### 5.1.2 Count

Regarding Count-1, the results look similar across all models. The accuracy starts at chance level and quickly increases. This is followed by a plateau (observed as constant brightness in Figure 4) during the first few layers of the LLM, until the middle layers, where there is a jump in accuracy within a few layers. For instance, for Ovis2.5 at 10 meters, the chance-corrected probe accuracy rises from 67.6% at layer 12 to approximately 87.8% at layer 16. This is then followed by a second plateau until the last layer. For Count-2, the pattern is quite similar. The probe's accuracy tends to be slightly higher across all models, which can be attributed to the fact that traffic barrels are always neatly aligned next to each other in the images, in contrast to pedestrians, which can overlap. Apart from this, we observe that the projector constitutes a small bottleneck for Ovis2.5 and InternVL3.5, which, however, does not appear to negatively affect the representation quality later in the model, as performance recovers and further improves within the LLM.

Overall, from these results, we can conclude that the concept of the quantity of a specific object for small counts is well, though not perfectly, encoded in the vision encoder for short distances and less so for longer distances. This representation is later almost uniformly improved across all distances in the middle layers of the LLM, with Ovis2.5 and Qwen3-VL showing a small overall advantage compared to the other models in terms of representation quality.

### 5.1.3 Spatial relationship

We evaluated the representation quality for spatial visual concepts at two levels. In Figure 4, the results show whether the visual concept at hand is **explicitly** encoded in the activation space of the model, while in Figure 5, the results indicate whether the model's activations retain sufficient spatial structure to allow the correct answer to be inferred at a later stage.

Regarding Spatial-1, Figure 4 shows that the chance-corrected accuracy of the probes trained on the average-pooled activations remains close to zero within the vision encoder. It then exhibits a sharp increase, for short distances, within one or two middle layers of the LLM across all models, rising from roughly 10% to 90%, with DriveFusionQA showing a slightly lower increase. At the same time, the picture in Figure 5, based on probes trained on region-pooled activations, is slightly different. For Ovis2.5 and Qwen3-VL and 5-meter samples, for example, the chance-corrected accuracy reaches around 85% and 73%, respectively, indicating good preservation of spatial structure, sufficient to infer the correct answer. For the remaining models, some degree of linear separability is still observed for short distances of 5 and 10 meters, although the results are less satisfactory.

Taken together, these findings suggest that the vision encoder does not explicitly encode the spatial relationship between the active and inactive blinkers in its activation space. However, it appears to retain sufficient spatial structure such that, at a later stage, the LLM, given the relevant cues, can infer the correct answer and explicitly encode it in a linearly separable manner in its activations. This process likely gives rise to the spike in accuracy observed in the middle layers of the LLM. For longer distances of 20 meters or more, it appears that insufficient information is retained in the vision encoder, preventing the LLM from recovering the spatial relationship between the blinkers.

Regarding Spatial-2, we observe a somewhat different pattern. The concept is explicitly encoded to some extent within the vision encoder for short distances, especially for Ovis2.5, which reaches a maximum chance-corrected accuracy of approximately 76% in Figure 4. A similar spike in accuracy is then observed in the middle layers of the LLM, as in Spatial-1, but this time even for longer distances. For probes trained on region-pooled activations, the accuracy within the vision encoder is substantially higher, reaching near-perfect chance-corrected accuracy for short distances and remaining high throughout the rest of the model. Overall, these results indicate that the side of the road on which the pedestrian is walking can be explicitly and linearly encoded within the vision encoder to a non-perfect degree for short distances. At the same time, it is almost perfectly encoded implicitly through the preserved spatial structure in the vision encoder's activations.

When comparing Spatial-1 and Spatial-2, the large difference in performance, particularly for longer distances, can likely be attributed to two factors. First, a pedestrian is substantially larger than a blinker,

making it easier to detect and localise at greater distances. Second, pedestrians are far more common in the training data of these models, which likely leads to stronger and more robust representations.

### 5.1.4 Orientation

Figure 4 shows that the results for the visual concept of object orientation differ substantially from the previous cases. Specifically, for Orientation-1, there are no indications that the concept is explicitly and linearly encoded to a satisfactory degree at any stage of the architecture, as indicated by the low probe accuracy throughout the architecture. In contrast, when examining the region-pooled activations (Figure 5), orientation appears to be implicitly encoded to some extent in the middle layers of the vision encoder for 5-meter samples, particularly for Ovis2.5 and Qwen3-VL. However, the performance of the linear probes degrades across the remaining layers of the architecture.

For Orientation-2, the results are slightly better at short distances. In this case, the orientation of the bicycle appears to be explicitly encoded to some extent at shorter ranges. With respect to the region-pooled activations, we observe strong linear separability at short distances, especially for Ovis2.5 and Qwen3-VL, indicating that the vision encoder captures sufficient information to infer the bicycle's orientation. However, at longer distances of 20 meters or more, the chance-corrected accuracy approaches zero, suggesting an absence of spatial structure capable of encoding orientation. The larger object size likely explains the more favourable results compared to Orientation-1.

Overall, when considering the results in Figures 4 and 5 together, we observe a different pattern from that seen in the spatial relationship tasks. In this case, the LLM appears unable to infer an object's orientation and subsequently encode it linearly in its activations, even when this information seems to be implicitly encoded by the vision encoder. This is evident, for example, in Ovis2.5 on samples at 5 m and 10 m, where we see high probe accuracy in the vision encoder layers in Figure 5 but low probe accuracy in the LLM layers in Figure 4.

### 5.1.5 Statistical Analysis

Given the relatively limited size of the test set on which the results in Figures 4 and 5 are based, we conducted an additional statistical analysis to increase confidence in the findings. More specifically, for each reported average test accuracy (i.e., each individual square in Figures 4 and 5), we computed the lower bound of the 95% confidence interval using the Wilson Score Interval (Wilson, 1927). The corresponding results are reported in Appendix B in a format similar to Figures 4 and 5. As can be seen, even under the worst-case lower-bound estimates, the overall patterns discussed above remain largely unchanged.

### 5.1.6 Discussion

Overall, we observe that some coarse-grained visual concepts, such as *presence*, are explicitly and linearly encoded to a considerable extent within the vision encoder, whereas more fine-grained concepts, such as *count*, are encoded less effectively. For concepts that require spatial understanding, such as *spatial relationship* and *orientation*, the vision encoder appears to encode them only weakly, if at all, in its activation space. However, the vision encoder seems to implicitly preserve sufficient spatial structure for some of these concepts to be inferred downstream. This is evident in the spatial relationship tasks, where we observe accuracy spikes in the middle LLM layers (Figure 4), but not in the orientation tasks, which suggests poor grounding of this concept in visual cues.

Considering the combined results from Figures 4 and 5, we draw the following conclusions. First, the LLM appears to be the main bottleneck for orientation understanding at short distances. Although the vision encoder retains sufficient spatial structure, the LLM fails to explicitly and linearly encode this information in the activation vector of the final token in the sequence, which is necessary for successful task performance. Second, the vision encoder seems to be the primary bottleneck for the poor encoding of visual concepts at long distances, as this is where the performance gap between short and long distances emerges. At the same time, it is unreasonable to expect a vision encoder to represent a pedestrian equally well at 5 meters and 50 meters, particularly when it has no prior knowledge of the question these features will be used to

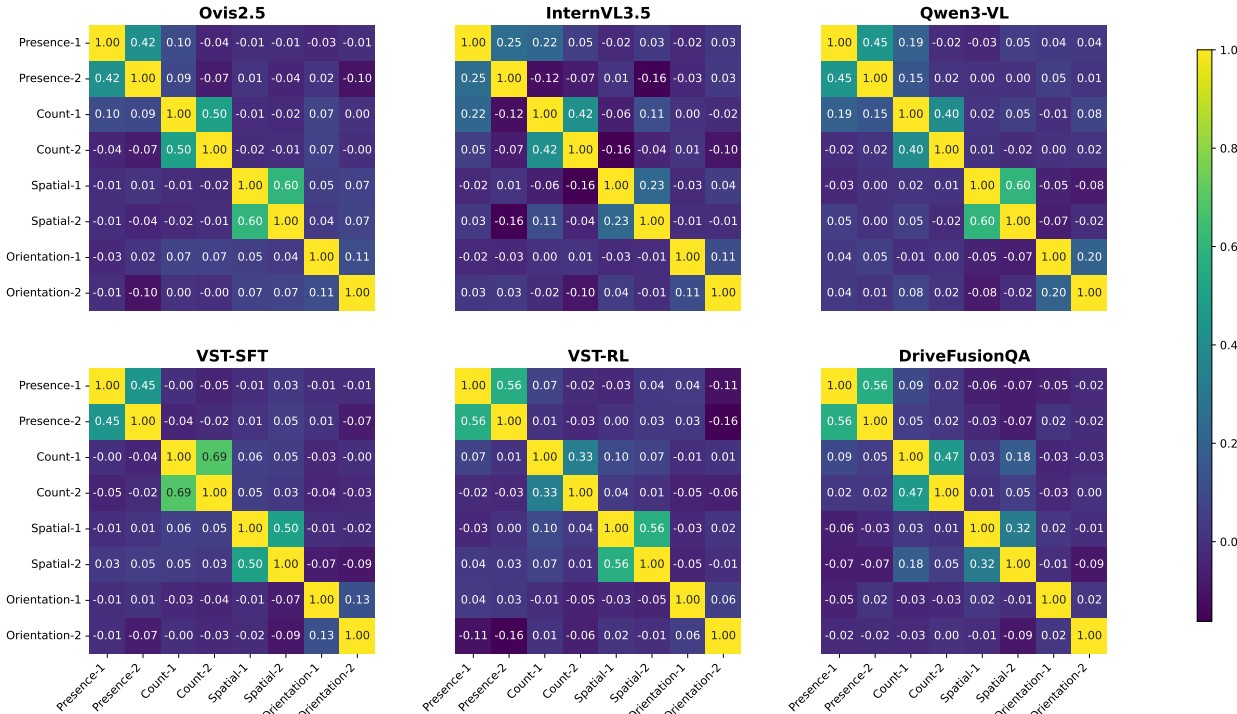

Figure 6: **Cosine similarity between the learned weights of the probes for each data category**. The probes considered here were trained on average-pooled activations of the last layer of the model (after layer normalisation) for the shortest-distance samples.

answer. Therefore, part of the failure at longer distances can also be attributed to the LLM, which does not sufficiently compensate for degraded visual representations.

Regarding the comparison among the six tested models, they exhibit broadly similar patterns, with several notable differences. First, in InternVL3.5, the projector acts as a strong bottleneck for *orientation*, whereas this effect is not present in the other models. Second, the Ovis2.5 vision encoder consistently performs substantially better across all tasks than the other models, with the largest gains observed in the spatial relationship and orientation tasks, particularly when using region-pooled activations. Finally, the two VST models and DriveFusionQA broadly exhibit the same patterns as the other models, indicating that fine-tuning and RL training on spatial or autonomous driving tasks do not materially change the intermediate visual concept encoding patterns.

Finally, an interesting observation is that, by examining the layer-by-layer flow of visual information in Figures 4 and 5, we are largely unable to distinguish among the three different VLM components. This suggests that the model effectively operates as a single, unified system, highlighting the importance of analysing the entire architecture end-to-end when seeking to understand its behaviour, rather than focusing solely on individual components such as the LLM.

## 5.2 Validating Learned Directions

In this section, we present a series of experiments that validate that the learned directions of the probes actually correspond to the target visual concept rather than merely capturing task-specific details for each data category, and that these directions causally influence the model's outputs.

### 5.2.1 Cosine Similarity

To assess whether the linear probes learn the direction of the target visual concepts in the models' activation space, we calculate the cosine similarity between the probe weights for all eight category types. This allows us to examine whether categories that represent the same visual concept exhibit high cosine similarity, as expected if the probes capture the underlying direction of the shared concept. We present the results in Figure 6. For each model, we use the probes trained on the average-pooled activations of the last layer (after layer normalisation), using the shortest-distance samples. Additionally, we consider only the second half of the learned weight vector, corresponding to the activations of the last token in the sequence (see §4.3). For Count-1 and Count-2, the probes learn a weight matrix $\mathbf{W} \in \mathbb{R}^{5 \times 4096}$ corresponding to different object counts. To represent the count concept as a single vector, and thus enable the computation of cosine similarity, we extract a difference vector $v = w_2 - w_1$, which captures the directional encoding in the activation space that shifts the representation from one object to two objects.

Looking at Figure 6, we observe a 2×2 block-diagonal structure, which indicates that the learned directions for data categories representing the same visual concept are more similar to each other than to the rest of the categories. This shows that the probes capture the underlying visual concept to some extent and not just the specifics of each data category. Of course, we must remember that the data we use are only proxy tasks for the underlying concept, and hence we cannot expect the learned probe weights to be identical, that is, to have a cosine similarity equal to 1, for data categories that belong to the same visual concept.

Regarding Orientation-1 and Orientation-2, we see that the learned directions of the probes are less similar, achieving a cosine similarity of up to 0.20 across all models. This is, in most cases, still higher than the similarity with data representing different visual concepts, but considerably lower than that observed for pairs of other visual concepts. This is consistent with the results in §5.1, which showed that the probes were unable to distinguish between the two orientations to a good degree, and therefore they definitely did not capture the underlying visual concept. Additionally, we observe that the level of similarity between the two *presence* categories and the two *spatial* categories for InternVL3.5 is much lower, although it still retains the 2×2 block structure, despite the good performance of the probes in §5.1. We speculate that InternVL3.5 encodes more specific details rather than the general underlying concept, for example, focusing more on the identity of the object present in the scene rather than on the general concept that something is present.

### 5.2.2 Activation Steering

Here, we use the learned weights of the probes as steering vectors (Turner et al., 2023; Zou et al., 2023; Li et al., 2023b) during generation. Our goal is to determine whether the directions in the models' activation spaces learned by the probes are actually used by the model during generation, that is, whether they are causal or merely correlational. More specifically, given an input image and the prompt "Describe the image briefly." we first record the model's answer. Then, using the same image and prompt as inputs, we steer the model's activations at a specific layer towards the learned direction of the probes; that is, we add or subtract the probe weights from the activations (with an adequate scaling factor) and record the resulting output. We then compare this output to the original one. For this experiment, we used probes trained on average-pooled activations. More details on the application of activation steering along with quantitative results are provided in Appendix C. We present some representative results in Table 3.

As we can see, across all models and data categories, steering using the probes' weights can yield the expected effect, that is, the model changes its answer based on how the encoded visual concept changed due to steering the activations. An exception to this is the orientation data, and hence we don't present results from these categories in Table 3. Moreover, despite intervening in the models' activations, we do not appear to break their behaviour. The rest of the scene description remains accurate, and the primary change concerns only the targeted visual concept. This supports the claim that these concepts are not only encoded linearly but also orthogonally to other concepts within the model (Park et al., 2024).

Table 3: **Activation steering results**. Each row shows how applying the directions learned by the linear probes as steering vectors alters the model's output, changing the original description to the steered description. The "Steering" column specifies the layer and token positions where the steering vector is applied. Additional details along with quantitative results are provided in Appendix C.

| Model | Input Image | Original Description | Steering | Steered Description |
|---|---|---|---|---|
| Ovis2.5 |  | The image shows a **person** walking across a road in a rural area with a red barn, cornfields, and a stop sign. The sky is overcast, and there are utility poles and streetlights along the road. | person → no person (LLM Visual Embeddings-Layer 1) | The image shows a road lined with cornfields on both sides, with a red barn on the left and a yield sign on the right. The sky is overcast, and the cornfields are tall and green. |
| InternVL3.5 |  | The image depicts a rural road scene with a windmill and a red barn on the left side. The road is flanked by tall cornfields on the right, with a stop sign visible in the distance. A **traffic barrel** is placed in the middle of the road... | traffic barrel → no traffic barrel (LLM Visual Embeddings-Layer 1) | The image depicts a rural landscape featuring a windmill on the left, a red barn, and a field of tall green crops on the right. In the foreground, there is a road with a barrier and a fence running alongside it. The sky is overcast... |
| Qwen3-VL |  | This is a screenshot from a video game, likely a simulation or open-world title, depicting a rainy, rural scene. The perspective is from inside a vehicle, looking out at a wet road. **A man** in a blue shirt and khaki pants is crossing the road... | one person → two people (LLM Visual Embeddings-Layer 1) | This is a screenshot from a video game, likely "The Sims" or a similar simulation game, showing a rainy, outdoor scene. The game features **two characters, a man and a woman**, walking on a wet road... |
| VST-SFT |  | The image depicts a city street scene with **a prominent orange traffic barrel** in the center. Tall buildings, including a skyscraper, line the background. The street is empty, and there are trees and streetlights along the sidewalks. | one traffic barrel → two traffic barrels (LLM Visual Embeddings-Layer 1) | The image depicts a city street scene with a focus on **a pair of orange traffic cones** placed in the center of the road. The street is lined with buildings on both sides, including a modern high-rise in the background... |
| VST-RL |  | The image shows a red truck with its rear lights on, positioned on a wet city street at night. **The left blinker is illuminated.** | left blinker → right blinker (LLM Visual Embeddings and Last Token-Layer 27) | The image shows a red truck with a flatbed driving on a wet city street at night. **The right blinker is illuminated.** |
| DriveFusionQA |  | The pedestrian is walking on the **left** side of the road. | left side → right side (LLM Visual Embeddings and Last Token-Layer 27) | The pedestrian is walking on the **right** sidewalk. |

### 5.2.3 Out of Distribution Evaluation

Finally, we evaluate the probes on out-of-distribution data to assess whether they generalise. The underlying idea is that, if a model truly dedicates a direction in its activation space to a specific visual concept, then the probe should be able to detect this concept even when the original traffic scene image is not from CARLA. More specifically, we use data from the Distance-Annotated Traffic Perception Question Answering (DTPQA) benchmark (Theodoridis et al., 2025a)[10] that are equivalent to the Presence-1 and Count-1 data but are drawn from nuScenes (Caesar et al., 2020). All data depict pedestrians at approximately 5 meters (2.5 to 7.5 meters), and therefore, we use probes trained on 5-meter CARLA data. These two data categories are the only ones for which nuScenes annotations allow us to automatically find equivalent images. Orientation-1 is also supported by the nuScenes annotations, but linear probes showed poor performance for this concept even in-distribution, so we considered it redundant to evaluate it out-of-distribution. To distinguish these datasets from the CARLA data, we refer to them as Presence-Real and Count-Real, respectively.

We present the results of this evaluation in Table 4, where we can see that the linear probes generalise extremely well in most cases. These results show that VLMs appear to consistently represent at least these two visual concepts across different types of data and confirm once again that the probes indeed learn a feature direction corresponding to the underlying visual concept.

### 5.3 Perceptual and Cognitive Failure

When evaluating the models on the same counterfactual images, as described in §4.2 (detailed results for the shortest-distance samples can be found in Appendix A), we observe cases where the models underperform compared to the linear probe trained on the average-pooled activations of the model's last layer (after layer normalisation). Figure 7 visualises this accuracy gap, defined as $probe\_acc - model\_acc$.

---

[10]This paper describes the dataset used in Theodoridis et al. (2025b), with substantial overlap between the two papers.

Table 4: **Probes accuracy on nuScenes data**.

| Model | Presence-Real Acc. (%) | Count-Real Acc. (%) |
|---|---|---|
| **Ovis2.5** | 82.1 (138/168) | 84.6 (44/52) |
| **InternVL3.5** | 81.5 (137/168) | 78.9 (41/52) |
| **Qwen3-VL** | 89.9 (151/168) | 61.5 (32/52) |
| **VST-SFT** | 97.6 (164/168) | 90.4 (47/52) |
| **VST-RL** | 97.0 (163/168) | 75.0 (39/52) |
| **DriveFusionQA** | 97.6 (164/168) | 82.7 (43/52) |

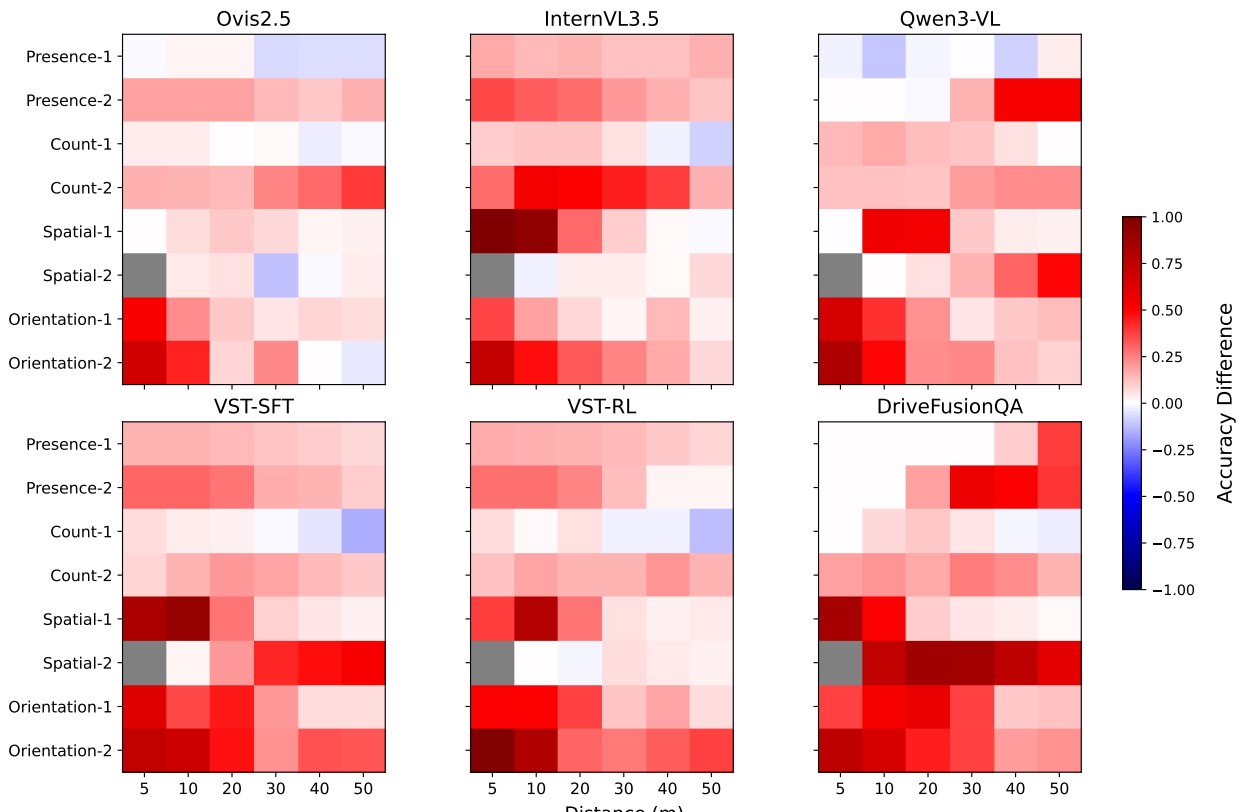

Figure 7: **Chance-corrected accuracy gap**. The gap is calculated between the test accuracy of the probes trained on the last-token activations of the last layer of the model (after layer normalisation) and the test accuracy of the models themselves. Grey indicates the absence of Spatial-2 data at 5 meters.

For this experiment, we trained new linear probes, this time using only the activations of the last token in the sequence, and measured their accuracy on the test set. This was done to ensure that the generative accuracy of the models and the linear probe accuracy rely on the same latent representation. If the absence of linearly encoded visual information were the sole reason for model failure, we would expect similar accuracy between models and probes.

However, we observe that in many cases the probe substantially outperforms the model. In some instances, this gap is large, with the model's accuracy close to chance while the probe's accuracy is close to perfect, as in the case of Spatial-1 at 5 meters for InternVL3.5. That means that in many cases, even though the visual information is linearly encoded at the last layer, the model still fails to make use of it and answer the question correctly; that is, it fails to assign the highest probability to the token of the correct answer.

Based on this result, we believe it is useful to think of two modes of failure. The first is **perceptual failure**, when the visual information is not linearly encoded at the last layer of the model, resulting in both low probe accuracy and low model accuracy. The second is **cognitive failure**, when the visual information is encoded at the last layer of the model but the model fails to correctly align this information with the language space, and hence to map this information to a probability distribution that favours the correct token. This corresponds to low model accuracy despite high probe accuracy.

Looking at Figure 7, we notice that for some models, the gap between probe and model performance is overall bigger compared to others. More specifically, Ovis2.5 seems less susceptible to cognitive failure compared to the rest of the models. Additionally, we observe that there are specific data types where cognitive failure seems to occur more often, for example, Count-2, Spatial-1 (apart from Ovis2.5), and Orientation-2. We include some early exploration of potential reasons behind cognitive failure in Appendix D.

We believe it is useful to think in terms of two distinct failure modes, because they lead to the same outcome, namely the model failing to answer a simple visual question, but arise from different causes and therefore require different remedies. For example, a perceptual failure might be addressed by improving the vision encoder, while a cognitive failure might be addressed by refining the training strategy of these models, particularly how visual features are aligned with the language space of the LLM component.

## 6    Limitations and Future Work

The primary limitation of our study is its dependence on counterfactual image sets, which are difficult to acquire in large numbers. Consequently, we limited the study to four visual concepts, with each represented by only two data categories. A straightforward future direction would be to scale up the current work by studying more visual concepts and representing each with more data categories. Another limitation of this paper is that it is based only on synthetic data, as it is difficult to obtain identical real traffic scenes at scale, where the only difference is the targeted visual concept. CARLA produces images with simplified texture statistics, more uniform material properties, and a narrower distribution of object appearances compared to real-world scenes (Pasios & Nikolaidis, 2025). As a result, we cannot be certain that the visual concepts we find to be encoded in the models would also be encoded in the same way when using real images. The out-of-distribution evaluation, the results of which are presented in Table 4, partially addresses this concern for `presence` and `count`, but the same is not true for `spatial-relationship` and `orientation`. Therefore, another direction for future work would be to include real data for all data categories. A further limitation is that our experiments were restricted to linear probes. Future work could explore non-linear probes and compare the results. It would also be interesting to repeat this study for larger VLMs (i.e., more than 4 billion parameters) to see to what degree the patterns observed for small models persist. Finally, while our work provides insights into how specific visual concepts are encoded and identifies distinct failure modes, it does not propose methods to address these limitations. An important direction for future research would therefore be to leverage the understanding of model internals provided by this work to develop techniques that mitigate the limitations and failure modes identified in our analysis.

## 7    Conclusion

In this work, we aim to improve our understanding of why SOTA small VLMs often fail on seemingly trivial visual tasks that are nevertheless crucial for interpreting traffic scenes. By applying linear probes to the intermediate activations of the models, we identify specific bottlenecks in the flow of visual information through the architecture for four visual concepts: presence, count, spatial relationship, and orientation. Our results show that, while basic visual concepts are explicitly and linearly encoded in the model, more fine-grained concepts are not, especially at greater distances. This is especially concerning for automated driving applications, where traffic scenes contain both coarse and fine-grained visual elements at varying distances. Furthermore, we identify two distinct failure mechanisms, perceptual and cognitive failure, which have different underlying causes and therefore may require entirely different mitigation strategies. We argue that interpretability research of this kind, focused on specific downstream tasks like understanding a traffic

scene, is essential for guiding the adoption of general-purpose VLMs in specialised domains, like automated driving.

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

# A Detailed Results

In Table 5, we present detailed results for all models and data categories at the shortest distance, together with the performance of probes trained on activations from the last vision encoder and LLM layers for comparison. Additionally, we report evaluation results obtained using a decoding strategy different from the greedy decoding described in §4.2. Specifically, instead of embedding the question in a prompt that instructs the model to answer with a single word, we directly ask the question without providing the possible answers and compare the probabilities assigned by the model to each potential answer. We refer to this decoding technique as constrained decoding. Overall, constrained decoding yields poorer results, which is why we rely on the first method; however, we report these results here for completeness. Prompt sensitivity is a well-known issue, but we do not explore it further as it falls outside the scope of this paper.

Table 5: **Detailed Evaluation Results**. We report the accuracy achieved by each model on the test set (Gen.) when prompted with images in which the object of interest is at a distance of 5 meters (or 10 meters for Spatial-2). Additionally, we report the accuracy obtained under constrained generation (Constr. Gen.). Finally, we report the accuracy of linear probes trained on the models' activations at the last vision encoder layer (Vis. Enc.) and at the last LLM layer (LLM), based on the activations of the last token in the sequence only.

| Model | Category | Accuracy (%) | | | |
|---|---|---|---|---|---|
| | | Gen. | Constr. Gen. | Vis. Enc. | LLM |
| Ovis2.5 | Presence-1 | 99.0 | 96.0 | 94.9 | 99.5 |
| | Presence-2 | 91.0 | 81.0 | 99.1 | 100.0 |
| | Count-1 | 86.5 | 87.0 | 72.8 | 88.9 |
| | Count-2 | 86.8 | 86.0 | 94.6 | 98.4 |
| | Spatial-1 | 100.0 | 50.0 | 57.7 | 100.0 |
| | Spatial-2 | 98.0 | 58.0 | 62.6 | 99.8 |
| | Orientation-1 | 50.0 | 49.0 | 66.4 | 64.9 |
| | Orientation-2 | 49.0 | 55.0 | 86.2 | 84.0 |
| InternVL3.5 | Presence-1 | 87.0 | 82.0 | 98.1 | 94.8 |
| | Presence-2 | 82.0 | 77.0 | 93.8 | 99.6 |
| | Count-1 | 75.8 | 34.8 | 61.3 | 84.6 |
| | Count-2 | 74.2 | 45.8 | 73.8 | 96.1 |
| | Spatial-1 | 50.0 | 50.0 | 53.9 | 99.9 |
| | Spatial-2 | 98.0 | 74.0 | 72.6 | 97.9 |
| | Orientation-1 | 53.0 | 50.0 | 56.2 | 63.0 |
| | Orientation-2 | 50.0 | 50.0 | 70.1 | 79.0 |
| Qwen3-VL | Presence-1 | 94.0 | 87.0 | 99.8 | 95.7 |
| | Presence-2 | 100.0 | 96.0 | 96.3 | 100.0 |
| | Count-1 | 78.3 | 70.3 | 74.4 | 88.2 |
| | Count-2 | 88.0 | 72.8 | 87.0 | 97.6 |
| | Spatial-1 | 100.0 | 52.0 | 52.8 | 99.9 |
| | Spatial-2 | 100.0 | 50.0 | 73.6 | 100.0 |
| | Orientation-1 | 45.0 | 49.0 | 58.0 | 60.5 |
| | Orientation-2 | 40.0 | 48.0 | 80.1 | 75.0 |
| VST-SFT | Presence-1 | 92.0 | 75.0 | 99.5 | 99.7 |
| | Presence-2 | 85.0 | 56.0 | 96.8 | 100.0 |
| | Count-1 | 82.5 | 43.8 | 70.0 | 87.8 |
| | Count-2 | 90.5 | 47.8 | 80.2 | 96.6 |
| | Spatial-1 | 58.0 | 50.0 | 55.7 | 98.1 |
| | Spatial-2 | 99.0 | 65.0 | 67.2 | 99.9 |
| | Orientation-1 | 50.0 | 50.0 | 67.4 | 62.6 |
| | Orientation-2 | 49.0 | 50.0 | 81.5 | 85.5 |
| VST-RL | Presence-1 | 91.0 | 86.0 | 99.5 | 99.9 |
| | Presence-2 | 86.0 | 78.0 | 96.8 | 100.0 |
| | Count-1 | 81.5 | 48.5 | 66.3 | 88.0 |
| | Count-2 | 87.8 | 48.5 | 82.7 | 97.1 |
| | Spatial-1 | 81.0 | 50.0 | 55.9 | 99.7 |
| | Spatial-2 | 100.0 | 100.0 | 71.9 | 100.0 |
| | Orientation-1 | 50.0 | 50.0 | 70.8 | 66.7 |
| | Orientation-2 | 38.0 | 50.0 | 82.0 | 87.0 |
| DriveFusionQA | Presence-1 | 100.0 | 98.0 | 93.1 | 100.0 |
| | Presence-2 | 100.0 | 83.0 | 97.2 | 100.0 |
| | Count-1 | 83.4 | 33.0 | 63.4 | 82.2 |
| | Count-2 | 82.3 | 43.3 | 73.8 | 96.9 |
| | Spatial-1 | 50.0 | 53.0 | 55.2 | 89.2 |
| | Spatial-2 | 63.0 | 50.0 | 71.5 | 99.0 |
| | Orientation-1 | 50.0 | 50.0 | 68.6 | 66.6 |
| | Orientation-2 | 50.0 | 50.0 | 79.3 | 79.4 |

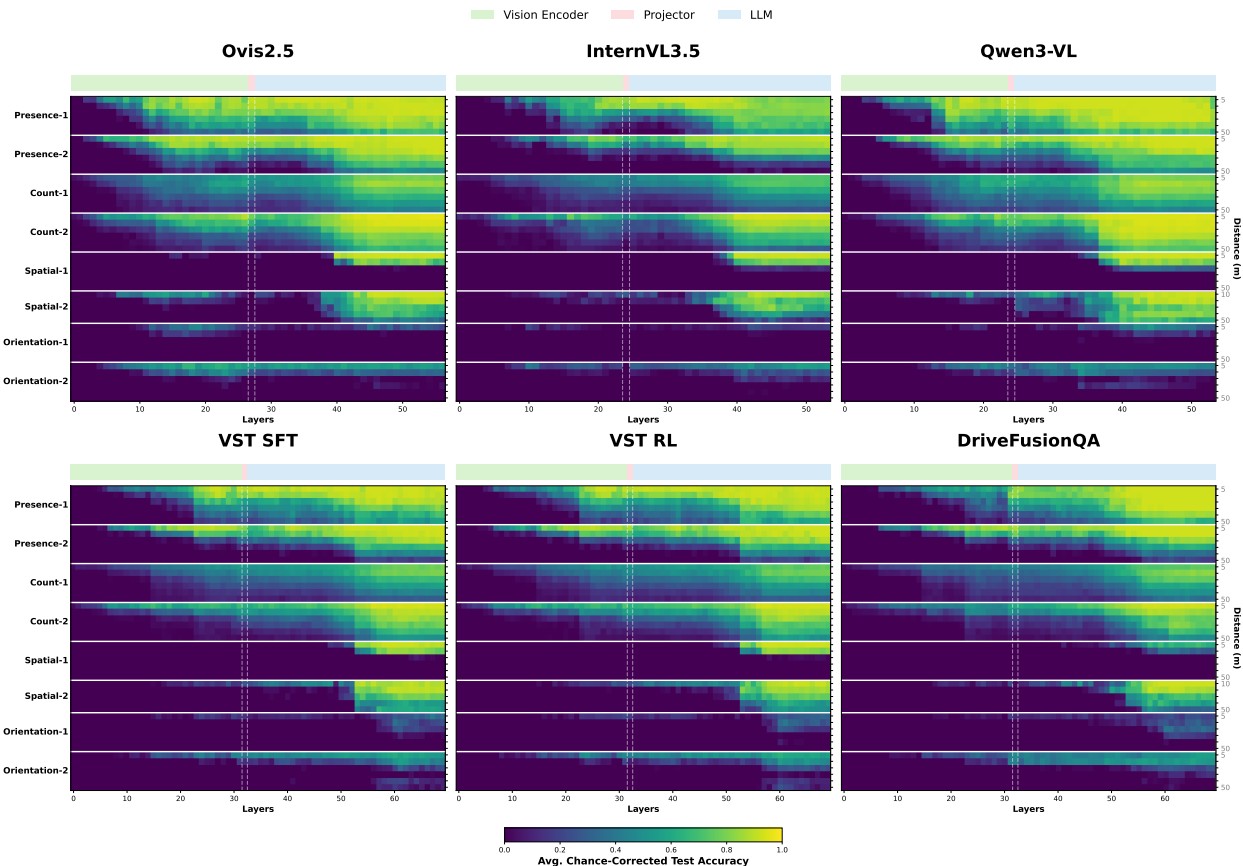

Figure 8: **Lower-bound equivalent of Figure 4**.

## B   Statistical Analysis

Here we present the results of the statistical analysis conducted to increase confidence in the results presented in Section 5.1 and to compensate for the relatively small size of the test set. As explained in Section 5.1.5, we calculated the lower bound of the 95% confidence Wilson score interval Wilson (1927) using the following equation:

$$\mathrm{WilsonLowerBound}(a_o, n, z) = \frac{a_o + \frac{z^2}{2n} - z\sqrt{\frac{a_o(1-a_o)}{n} + \frac{z^2}{4n^2}}}{1 + \frac{z^2}{n}} \tag{7}$$

where $a_o$ is the observed test accuracy, $n$ is the number of test samples, and $z$ is the z-score corresponding to the desired confidence level (for 95% confidence, $z = 1.96$).

Finally, to efficiently present the lower-bound values, we recreated the "lower-bound" versions of Figures 4 and 5. More specifically, for each combination of model, data category, distance, and layer, we first calculated the lower-bound accuracy and then computed the corresponding chance-corrected accuracy. The resulting plots are presented in Figures 8 and 9. Overall, the average difference between the observed accuracy and the lower-bound is 7.93 pp (percentage points) with a standard deviation of 2.13 pp for the results in Figure 8 and 8.87 pp with a standard deviation of 1.66 pp in Figure 9. These numbers correspond to the differences before correcting for chance accuracy.

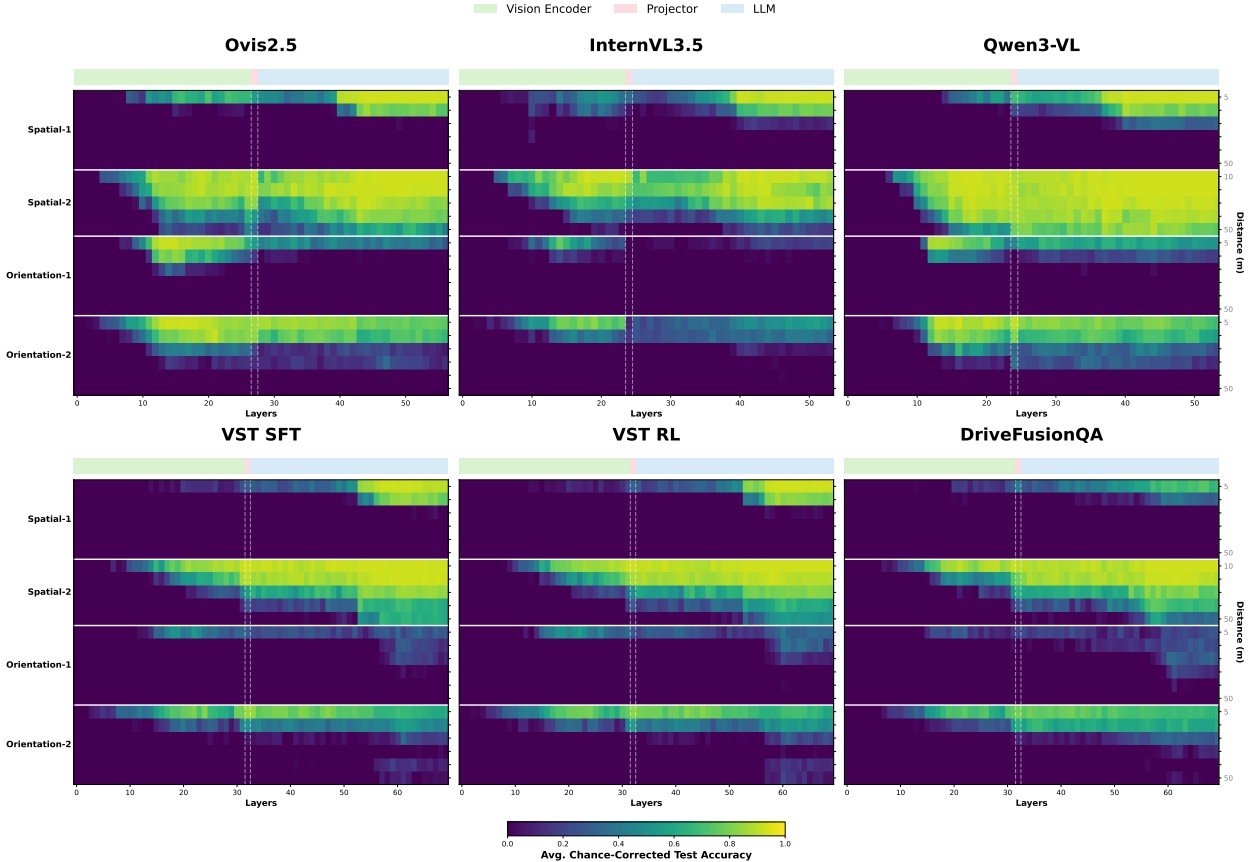

Figure 9: **Lower-bound equivalent of Figure 5**.

## C  Activation Steering

Here we explain the details of the activation steering experiment, the results of which are presented in Table 3. As mentioned before, the goal of this experiment was to determine whether the directions learned by the probes have a causal effect on the model's output. When we refer to the direction of a visual concept $c$, we mean the direction that moves the representation from one state $c_i$ to another state $c_j$ (e.g., from the state *left* of the visual concept *orientation* to the state *right*). For simplicity, we refer to this as the direction of concept $c$ without explicitly mentioning the states each time.

We first define the learned direction of the probes for a concept $c$ as the element-wise average of the learned weights, excluding the bias, of the best probes across all ten runs. For example, for layer $l$, the probe's learned direction is defined as:

$$\mathbf{w}^{(l)} = \frac{1}{10} \sum_{i=1}^{10} w_i^{(l)} \tag{8}$$

where $w_i^{(l)}$ denotes the learned weights, excluding the bias, of the $i$th probe. For probes trained on the *count* concept, where there are five output neurons, and hence the learned weights are $\mathbf{w} \in \mathbb{R}^{5 \times d_L}$, we choose to work with the direction that points from *one* to *two*, by taking the element-wise difference of the third dimension of $w$ with its second dimension (the ones corresponding to the answer *two* and *one* respectively).

However, it is important to note that the probes were trained on standardised activations, whereas we use the resulting directions as steering vectors in the raw activation space. Therefore, we divide $\mathbf{w}^{(l)}$ by the standard deviation of the activations used during standardisation, thereby mapping the learned direction back to the model's original activation space. Finally, because we want the steering vectors to have unit norm, we define the steering vector $\mathbf{s}^{(l)}$ as:

$$\mathbf{s}^{(l)} = \frac{\mathbf{w}^{(l)}/\boldsymbol{\sigma}}{\left\|\mathbf{w}^{(l)}/\boldsymbol{\sigma}\right\|_2} \tag{9}$$

We use as input to the model an image depicting concept $c_i$, together with the prompt "Describe this image briefly.", and save the generated text output. We found that, for DriveFusionQA, this prompt often resulted in very short responses such as "The car is driving forward", without providing a meaningful description of the scene. Therefore, for this model, we instead used the prompt suggested on its Hugging Face page (Samir & Team, 2026): "Describe the current driving scenario and any potential risks."

Additionally, we observed that, for Spatial-1, the models would rarely mention the blinker, as it is a relatively minor detail. To address this, we appended the instruction "State whether the left or right blinker is illuminated." to the prompt in order to encourage the model to describe this aspect of the scene. Similarly, for Spatial-2, while the models typically mentioned the pedestrian, they almost never specified which side of the road the pedestrian was located. Therefore, we added the instruction "Focus on where the pedestrian is walking (i.e. which side of the road)." to the prompt. We only used images where the object in question is in the closest proximity for this experiment.

Subsequently, we use $\mathbf{s}^{(l)}$ as a steering vector for the activations of the VLM at layer $l$ and provide the same input image and prompt to the model, while simultaneously steering the corresponding activations at layer $l$ from $c_i$ to $c_j$. By comparing the original generated text to the steered one, we can qualitatively assess whether the learned direction of the probes is actively used by the model when generating its output.

We applied steering vectors to LLM layers, although the same could be applied to the vision encoder and projector parts as well. There are three possibilities when steering the activations of a specific layer $l$ within the LLM component, given that the learned weights of the probes are based on the activations of the visual tokens and the last token in the sequence (§4.3). We can either steer the activations of the visual tokens, the activations of the last token in the sequence, or the activations of both. Steering the activations of the visual tokens uses the first half of $\mathbf{s}^{(l)}$, which is learned based on the visual tokens, while steering the last token activations uses the second half of $\mathbf{s}^{(l)}$, which is learned based on the last token. More formally, when steering the visual tokens at layer $l$, we obtain:

$$\tilde{\mathbf{l}}^{(l)}_{visual} = \mathbf{l}^{(l)}_{visual} + \alpha \cdot \mathbf{s}^{(l)}_{first} \tag{10}$$

where $\mathbf{s}^{(l)}_{first}$ denotes the first half of the steering vector and $\alpha$ is a scale factor. $\mathbf{s}^{(l)}_{first}$ is broadcast across the first dimension before being added to the existing activations. When steering the last token at layer $l$, we obtain:

$$\tilde{\mathbf{l}}^{(l)}_{t-1} = \mathbf{l}^{(l)}_{t-1} + \alpha \cdot \mathbf{s}^{(l)}_{second} \tag{11}$$

where $\mathbf{s}^{(l)}_{second}$ refers to the second half of the steering vector.

We performed a small grid search over $\alpha$, testing five different values per sample. In Table 3, we present the results obtained using the smallest absolute value of $\alpha$ that caused a semantic change in the model's description of the scene.

It is important to note that we did not perform an extensive exploration of other steering-related hyperparameters, such as which model component or layer is most effective for steering, or whether steering should be applied only to the visual tokens, only to the last token, or to both within the LLM. Determining the optimal steering strategy is outside the scope of this experiment. Instead, we applied steering at the earliest LLM layer where probe accuracy was high. When steering was applied at the first layer, we modified only

| Model | Presence-1 | Presence-2 | Count-1 | Count-2 | Spatial-1 | Spatial-2 |
|---|---|---|---|---|---|---|
| Ovis2.5 | 94.0 (47/50) | 100.0 (50/50) | 93.8 (45/48) | 60.0 (30/50) | 76.6 (36/47) | 100.0 (48/48) |
| InternVL3.5 | 100.0 (50/50) | 100.0 (48/48) | 75.0 (36/48) | 42.6 (20/47) | 83.9 (26/31) | 87.8 (43/49) |
| Qwen3-VL | 100.0 (50/50) | 100.0 (49/49) | 73.9 (34/46) | 71.4 (35/49) | 97.0 (32/33) | 100.0 (49/49) |
| VST-SFT | 100.0 (50/50) | 100.0 (46/46) | 97.9 (47/48) | 71.4 (35/49) | 97.4 (37/38) | 100.0 (49/49) |
| VST-RL | 100.0 (50/50) | 100.0 (38/38) | 78.3 (36/46) | 64.4 (29/45) | 100.0 (42/42) | 87.8 (43/49) |
| DriveFusionQA | 100.0 (40/40) | 100.0 (5/5) | 7.9 (3/38) | 0.0 (0/5) | 88.2 (15/17) | 100.0 (45/45) |

Table 6: **Activation steering success rate (%) by model and data category**. Percentages are computed over valid steering cases only; counts in parentheses indicate successful cases over valid cases.

the visual tokens, as this should theoretically be sufficient. In contrast, when steering at later layers for the Spatial-1 and Spatial-2 settings, we also steered the last token. At this stage, steering the visual tokens alone is insufficient, since the last token has already formed a consolidated representation of the scene.

We performed activation steering for each model and data category using 50 randomly selected samples, and report the success ratio in Table 6. More specifically, a success is defined as a sample where the base description ($\alpha = 0$) mentions the original concept state and at least one of the steered descriptions changes to the target concept state, while the rest of the description remains accurate. If the original concept state is not mentioned in the base description, the sample is considered undefined and is not included in the success-rate calculation. Finally, if the original concept state is mentioned in the base description but the target concept state is never mentioned in any of the steered descriptions, the sample is considered a failure.

We do not count undefined cases as failures because, if the model does not mention the original concept state in the first place, then the direction learned by the probes cannot have an observable effect on the generated description. Such samples, therefore, do not provide useful information about the causality of the probe directions.

As shown in Table 6, using the probe-learned directions to steer the models' responses achieves nearly 100% success rates across all models for the Presence categories, and very high success rates for the remaining categories. The only exception is DriveFusionQA. Even after applying its recommended prompt, the model often produces a generic response such as "The car is driving forward slowly", without providing additional scene details. This makes it impossible to reliably evaluate steering success, particularly for Presence-2 and Count-2, both of which involve traffic barrels in the scene.

For Count-1, however, 38 samples produced valid responses, of which only three were successfully steered. At the same time, Figure 4 shows that Count-1 information is well encoded in the model's activations. This suggests that, although the fine-tuning undergone by DriveFusionQA preserved the linear accessibility of count information, it may have weakened the causal influence of this information on the generated output. Consequently, intervening in the model's activations may cause it to fall back on a generic default response, such as "The car is driving forward slowly", which is what we observe. However, because the steering configuration was not exhaustively optimised, the present experiment cannot isolate fine-tuning as the cause.

## D   Further analysis of Cognitive Failure

In this section, we take a closer look at cognitive failure and its potential causes. More specifically, we again train a linear probe, but this time on the generated logits distributions, with the goal of identifying which tokens are most discriminative between counterfactual inputs. For all six models, the vocabulary size is approximately 150,000 tokens. Consequently, we require the probe to be sparse so that only a small number of non-zero weights are retained and can be meaningfully inspected. To this end, we train a logistic regression model on top of the logits distribution using an L1 penalty with $C = 0.3$, enforcing strong regularisation. We use only 24 samples per class for each data category. In addition, samples from all distances are included.

Our results show that, in many cases, attending to tokens that are semantically close to the correct and incorrect answers allows this simple logistic regression model to close, or nearly close, the gap between model

accuracy and probe accuracy. This suggests that a potential contributor to cognitive failure is the surface form competition (Holtzman et al., 2021). In particular, the vocabulary often contains many tokens that are semantically very similar to both the correct and incorrect answer tokens, and these tokens can effectively "steal" probability mass from the intended answer. When combined with greedy decoding, this redistribution of probability can lead to incorrect predictions.

Table 7 illustrates examples in which focusing on mostly semantically similar tokens enables the logistic regression model to close, or at least substantially reduce, the cognitive failure gap. For instance, in the first row of Table 7, multiple variations of the token "none" appear in the word cloud, alongside several variations of the token "yes." At the same time, non-semantically interpretable tokens also appear, such as a sequence of Chinese characters translating to "Deputy director." This indicates that the surface form competition is unlikely to be the sole cause of cognitive failure and instead is likely one contributing factor among several. Similarly, in the second row, where the correct answer is "Two," we observe that tokens such as "tw" or ".tw" may absorb sufficient probability mass to confuse the model. In these cases, the logistic regression probe is able to almost entirely close the observed cognitive gap by attending to such tokens.

Finally, Table 8 presents examples where this approach is insufficient to explain cognitive failure, underscoring the need for further investigation. In the first two rows, the logistic regression model effectively "gives up" by zeroing all its weights, indicating that it is not possible to improve accuracy by attending to a small number of tokens. In contrast, in the last two rows, the probe achieves a slight improvement over the original model, but the tokens it attends to are not semantically related to the expected answers. This suggests that, in these cases, the probe may be exploiting dataset-specific regularities, potentially overfitting to characteristics of the CARLA-generated data rather than capturing a genuine explanatory signal.

Table 7: **Logistic regression on logits results**. The word clouds show the tokens that correspond to non-zero weights. Increased size means a larger weight value. Blue and red colours mean positive and negative weight values, respectively.

| Model / Category | Positive/Negative answer(s) | Important Tokens | Translations | Accuracy Comparison |
|---|---|---|---|---|
| VST-RL / Presence-1 | Yes/No | | 想不到 = Unexpectedly
不开 = Not open
部副 = Debuty director | |
| Ovis2.5 / Count-1 | Two/Zero, One, Three, Four | | 二线 = Second-tier
三是 = Thirdly
三条 = Three | |
| InternVL3.5 / Count-2 | Two/Zero, One, Three, Four | | 前三 = Top three
双双 = Together
四 = Four | |
| Ovis2.5 / Presence-2 | Yes/No | | 都没有 = None whatsover
没有任何 = Nothing at all
我没有 = I haven't | |
| Qwen3-VL / Spatial-2 | Left/Right | | 右侧 = Right side
右边 = On the right
右 = Right | |
| DriveFusionQA / Spatial-2 | Left/Right | | 右侧 = Right side
删除 = Delete
右边 = On the right | |

Table 8: **Logistic regression on logits results**. The word clouds show the tokens that correspond to non-zero weights. Increased size means a larger weight value. Blue and red colours mean positive and negative weight values, respectively.

| Model / Category | Positive/Negative answer(s) | Important Tokens | Translations | Accuracy Comparison |
|---|---|---|---|---|
| Ovis2.5 / Orientation-1 | Right/Left | - | - |  |
| VST-SFT / Orientation-2 | Right/Left | - | - |  |
| VST-RL / Orientation-2 | Right/Left | Forward no 旋转 | 旋转 = Rotation |  |
| InternVL3.5 / Spatial-1 | Right/Left | CM front [P | - |  |

