# OpenReview forum: "Probing Visual Concepts in Lightweight Vision-Language Models for Automated Driving"
_TMLR — Decision pending for TMLR_

### Review · Reviewer_8qgi · 2026-03-24

**Summary Of Contributions:**

This paper investigates the internal mechanisms by which small Vision Language Models process visual information relevant to automated driving. The authors construct counterfactual image sets using the CARLA simulator, where each set of images is identical in all respects except for a single targeted visual concept: object presence, object count, spatial relationship, or object orientation. Linear probes are then trained on intermediate activations extracted from every transformer block in the vision encoder, the projector module, and the LLM of four model configurations (Ovis2.5-2B, InternVL3.5-2B, VST-3B in both its SFT and RL variants). Probe accuracy across layers is used as a proxy for linear encoding quality of each visual concept at each stage of the architecture.

The paper draws three principal conclusions:
- (1) coarse-grained concepts such as object presence are explicitly and linearly encoded beginning in the middle layers of the vision encoder and are preserved or improved throughout the LLM, while fine-grained spatial concepts such as object orientation are encoded only weakly and only implicitly through the spatial structure of patch activations
- (2) two distinct failure modes are identified, termed perceptual failure (the concept is not linearly encoded at the final model layer) and cognitive failure (the concept is linearly encoded but the model nonetheless produces incorrect outputs)
- (3) increasing object distance rapidly degrades linear separability throughout the architecture, with degradation originating primarily in the vision encoder.

Supplementary experiments validate the learned probe directions via cosine similarity analysis, out-of-distribution evaluation on nuScenes data, and activation steering using probe weights as steering vectors.

The principal strengths of the submission are its rigorous experimental design centered on controlled counterfactual inputs, its unified layer-by-layer analysis spanning the complete VLM architecture rather than individual components in isolation, the transparency and reproducibility of its methodology, and the genuine relevance of its findings to the autonomous driving community. The principal limitations, discussed further below, are an insufficiently thorough engagement with closely related contemporaneous work that establishes overlapping concepts under different terminology, some overclaiming in the discussion of orientation results, and a preliminary cognitive failure analysis (Appendix C) whose scope should be better calibrated relative to the strength of the evidence.

**Additional Comments:**

The writing is clear and generally well-organized throughout. The methodology section is technically precise and the activation extraction strategies (average pooling versus region pooling) are well-motivated and carefully distinguished. The cosine similarity analysis in Figure 6 is a thoughtful addition to the validation suite, and the 2x2 block-diagonal structure it reveals provides a useful sanity check that the probes are capturing conceptual directions rather than task-specific artifacts.

**Audience:**

Yes

**Audience Explanation:**

The findings of this paper address two intersecting communities that are both active within TMLR's readership: researchers working on VLM interpretability and mechanistic understanding, and researchers applying machine learning to autonomous driving and safety-critical perception. For the interpretability community, the layer-by-layer probing methodology applied to the complete VLM pipeline (including the vision encoder, which is typically omitted from such analyses) surfaces a concrete and reproducible set of patterns distinguishing coarse from fine-grained visual concept encoding, and provides an operational diagnostic framework for decomposing failure modes. For the autonomous driving community, the finding that orientation and long-distance object representation are consistently poor across architectures, and that a spatially trained model does not measurably improve internal visual encoding of these concepts, has direct implications for deployment decisions and for the design of future training objectives.

While individual components of this work build on established techniques (linear probes, counterfactual evaluation, activation steering), their combination in this specific domain and at this level of architectural granularity constitutes a contribution that offers actionable insight to researchers in both communities. Under TMLR's evaluation framework, which explicitly does not require novelty of the studied method as a criterion for acceptance, the interest criterion is satisfied. The paper offers clear lessons about where improvements in small VLMs should be directed, and the CARLA-based counterfactual dataset approach is a reusable and extensible methodology for future work in this area.

**Broader Impact Concerns:**

No significant broader impact concerns are raised. The paper is an interpretability study of small VLMs and does not introduce models, training methods, or data that could be directly misused. The autonomous driving application context creates implicit safety implications, which the authors address appropriately by framing their findings as evidence for caution in deploying general purpose VLMs in safety critical settings, particularly for fine-grained spatial reasoning tasks and for objects at distance. The CARLA-generated dataset raises no privacy or data consent issues.

The paper does not include a dedicated broader impact statement, but the nature of the work does not appear to require one under current TMLR norms. If the editor requests one, the authors could briefly address the safety implications of deploying spatially limited VLMs in automated vehicles, particularly in light of the orientation and long-distance encoding failures documented here.

**Claims And Evidence:**

Yes

**Claims Explanation:**

Overall, the central claims of this submission are well supported by the experimental evidence presented. Each claim is assessed individually below.

__Claim 1: Presence and count are explicitly linearly encoded.. orientation is not.__ This is the paper's strongest claim and is convincingly supported. The heatmaps in Figures 4 and 5 provide a clear, consistent picture across all four model variants. Presence detection achieves near-perfect chance corrected accuracy from the middle layers of the vision encoder onward at short distances under average pooling, a finding that is replicated across Ovis2.5, InternVL3.5, VST-SFT, and VST-RL. Orientation, by contrast, remains near zero under average pooling throughout the entire architecture for pedestrians (Orientation-1), and shows only marginal separability at short distances for bicycles (Orientation-2). The region-pooled results in Figure 5 confirm that some spatial structure is retained in the vision encoder for orientation, but that this structure degrades in the projector and LLM rather than being amplified. This is a meaningful and actionable finding.

One qualification is warranted. The abstract and conclusion describe orientation as being "only implicitly encoded by the spatial structure retained by the vision encoder." However, the region-pooled results for Orientation-1 (pedestrian) are notably weak even at 5 meters, with chance-corrected accuracy reaching roughly 0.2 to 0.3 for Ovis2.5 and substantially lower for other models, and deteriorating to chance at 10 meters. The qualifier "implicitly" is therefore most accurate for Orientation-2 (bicycle) at short distances, but overstates the encoding quality for the pedestrian orientation task. The authors should revise the relevant language in the abstract and conclusion to reflect this nuance.


__Claim 2: Two failure modes exist (perceptual vs. cognitive failure).__
The evidence for distinguishing these two failure modes is credible and the framework is analytically useful. Figure 7 clearly demonstrates cases where probe accuracy at the final layer substantially exceeds model accuracy, indicating that the model encodes the information necessary to answer correctly but fails to use it. The most striking case is Spatial-1 for InternVL3.5 at 5 meters, where the probe achieves near-perfect accuracy while model accuracy remains at chance level. This finding is consistent with prior work by Fu et al. (2025, COLM Outstanding Paper), who showed through a different methodology (task-specific visual readout probes across the pipeline) that VLMs routinely underperform relative to the information available in their own activations. Comparison to this work is important and must be added. Similarly, Vompa et al. (2025) introduced the "linear separability ceiling" concept, which formalizes the same probe-model accuracy gap as a bound on generative performance. The paper cites Vompa et al. in the related work section but does not discuss this connection when presenting Figure 7, where it is most directly relevant.

The activation steering experiments in Section 5.2.2 and Table 3 provide additional causal validation by showing that intervening on the learned probe directions changes model outputs in the expected direction. These results are qualitatively convincing for presence, count, and spatial relationship concepts. The authors appropriately note that orientation steering does not produce reliable results, which is consistent with the weak encoding evidence. The limitation of Table 3, however, is that it presents only a curated selection of successful cases without quantifying the success rate across all tested images and conditions. The authors should report how many images were tested per concept and what proportion of steering interventions produced the expected semantic change versus partial changes versus no change.

__Claim 3: Distance degrades linear separability, primarily in the vision encoder.__
This claim is well supported by the distance-stratified heatmaps in Figures 4 and 5. The degradation pattern is visible and consistent: short-distance performance (5 meters) is substantially better than long-distance performance (50 meters) across all concepts and models, and the gap emerges in the vision encoder layers rather than in the LLM. The partial recovery observed in the LLM for presence tasks (e.g., 24% chance-corrected improvement for Ovis2.5 between first and last LLM layers at 50 meters) is discussed clearly and appropriately. The nuScenes out-of-distribution evaluation in Table 4 further supports the claim that the presence and count probes generalize beyond CARLA-generated images, lending credibility to the claim that these findings reflect genuine model properties rather than simulator-specific artifacts.


__Claim 4: VST (spatially tuned) models do not substantially outperform general-purpose models.__
This finding is implicit rather than explicitly foregrounded but constitutes a practically important result. The heatmaps show that VST-SFT and VST-RL exhibit broadly similar encoding patterns to Ovis2.5 and InternVL3.5, with no systematic advantage on orientation or spatial relationship tasks in terms of linear separability. This is an appropriate and well-supported conclusion given the evidence. The comparison between VST-SFT and VST-RL is a thoughtful inclusion, and the observation that reinforcement learning does not materially change the pattern of visual concept encoding is noteworthy.

**Requested Changes:**

__Critical__

The following changes are considered critical because they affect the accuracy of the paper's positioning within the existing literature, and because their absence creates the impression that key precedents were overlooked.
First, Fu et al. (2025), "Hidden in Plain Sight: VLMs Overlook Their Visual Representations" (COLM 2025 Outstanding Paper), must be discussed in the related work as a primary predecessor. That paper demonstrates using task-specific probe heads trained across the full VLM pipeline that visual information is routinely preserved throughout the architecture while generative performance falls substantially below what the representations could support, which is precisely the "cognitive failure" phenomenon studied here. The paper is currently cited in Section 2.4 but only in passing, without describing its core finding. Given the substantive overlap in conclusion, the authors must explicitly characterize how their methodology differs from Fu et al. (counterfactual isolation of specific driving-relevant concepts, standardized linear probes at every layer including the vision encoder, distance stratification) and why those differences yield additional insight. Simply noting that the paper "showed VLMs perform worse than their vision encoders" understates what Fu et al. demonstrate and risks appearing to minimize the overlap.

Second, Li et al. (2023), "Inference-Time Intervention: Eliciting Truthful Answers from a Language Model" (NeurIPS 2023), must be cited in the activation steering section. That paper explicitly proposes using linear probe weight vectors as intervention directions and tests this approach for steering model outputs, which is mechanically identical to the approach described in Section 5.2.2. Turner et al. (2023) and Zou et al. (2023) provide the broader steering framework, but Li et al. provide the specific precedent for probe-weight steering, and its omission from the references is a substantive gap.

Third, the claim in the abstract that the paper analyses "four state-of-the-art VLMs" is slightly misleading, since VST-SFT and VST-RL represent two training variants of the same underlying architecture rather than four architecturally distinct models. The abstract should clarify that three model families are evaluated, with VST examined in two training configurations (supervised fine-tuning only versus supervised fine-tuning followed by reinforcement learning), to avoid overcounting.

Fourth, the description of Theodoridis et al. in the references includes what appear to be two separate citations for works by the same group: one labeled as 2025a for the DTPQA benchmark and another labeled as 2025b for the IEEE OJVT paper on evaluating small VLMs (arXiv 2510.08352). These should be verified as genuinely distinct publications with separate content, and the citations should be written to make the distinction unambiguous. If substantial content overlaps between these two works, the relationship should be acknowledged.


__Non-Critical__

The authors are encouraged to address the following points in a revision, though they are not considered blocking.

The language describing orientation encoding should be made more precise in the abstract, introduction, and conclusion. The current framing implies that orientation is encoded implicitly through spatial structure across all conditions and all models. The region-pooled results for Orientation-1 (pedestrian) are too weak and too inconsistent to support this characterization at more than 5 meters or beyond Ovis2.5. The revised text should clarify that implicit encoding of orientation is model-dependent and distance-limited, with Ovis2.5 being the strongest case.

The activation steering experiments in Table 3 would benefit from a brief quantitative summary. A simple count of how many tested inputs per concept yielded the expected semantic change, a partial change, and no change would substantially improve the interpretability of these results and allow readers to assess how robustly the probe directions are used by each model.

The Appendix C analysis of cognitive failure causes is described by the authors themselves as "early exploration," yet it is presented at the same level of formality as the main results. The authors should either (a) expand this analysis with cleaner experimental controls and a larger sample size, thereby transforming it into a more conclusive finding, or (b) more explicitly frame it as a preliminary investigation and reduce its implied contribution weight in the paper. The surface form competition hypothesis is plausible and aligns with prior work (Holtzman et al. 2021), but the logistic regression evidence presented is too thin and the non-semantic token results (Table 7) suggest the analysis may be fitting dataset artifacts in some cases.

The paper would benefit from a brief discussion of the potential influence of CARLA's visual statistics on the results. Counterfactual CARLA images may produce more consistently structured activation patterns than real-world images, which could make presence and count concepts appear more linearly separable than they would be in naturalistic scenes. While the nuScenes out-of-distribution results in Table 4 partially address this concern for presence and count, the spatial relationship and orientation results have no equivalent real-world validation. Acknowledging this limitation explicitly in the limitations section, with a note on how future work could address it, would strengthen the paper.

The paper includes three models in Table 1 but the abstract mentions "four state-of-the-art VLMs." Apart from the clarification requested above, the rationale for selecting these three specific model families over other equally available sub-4B models (such as PaliGemma-3B, Phi-3.5-Vision, or MiniCPM-V) deserves a sentence or two of justification beyond performance ranking.
Vompa et al. (2025) should be discussed more specifically at the point in Section 5.3 where Figure 7 is introduced. That paper's linear separability ceiling construct is directly relevant to interpreting the probe-model accuracy gap, and the connection between cognitive failure and the ceiling framework would enrich the discussion.

---

> ### Author Response · Authors · 2026-06-05
>
> We sincerely thank the reviewer for the thorough review. We address each point below.
>
> ---
>
> ### Critical
>
> **1. Fu et al. (2025), "Hidden in Plain Sight: VLMs Overlook Their Visual Representations", must be discussed as a primary predecessor.**
>
> We expanded the discussion of Fu et al. in Section 2.4 and moved it to the end of the paragraph to emphasize it as a primary predecessor:
>
> > Similarly, Fu et al. (2025b) show that VLMs perform worse than their vision encoders on several vision-centric tasks, and that this gap is not due to degradation of visual information in the projector or LLM. Instead, they show that visual information is preserved and accessible across these components, highlighting that VLMs can fail even when the relevant visual information is encoded. This aligns to some degree with what we describe as cognitive failure in our work.
>
> Regarding how our methodology differs, we already discussed this in the final paragraph of Section 2.4. However, we expect that moving the discussion of Fu et al. to the end of the preceding paragraph will make the distinction between our work and theirs clearer.
>
> **2. Li et al. (2023), "Inference-Time Intervention: Eliciting Truthful Answers from a Language Model", must be cited in the activation steering section.**
>
> We now cite Li et al. (2023), "Inference-Time Intervention: Eliciting Truthful Answers from a Language Model" in section 5.2.2.
>
> **3. The claim that the paper analyses "four state-of-the-art VLMs" is slightly misleading.**
>
> We revised this claim and now explicitly state in the abstract that we use five models (after adding two additional models in our analysis after the request of reviewer 6Xnc), including two training variants of one model.
>
> Later, when we refer to six models again in the Introduction (fifth paragraph), we added a footnote clarifying this point:
>
> > We actually use five distinct models with two training variants for one of them.
>
> This allows us to use the number "six" throughout the rest of the paper while avoiding confusion.
>
> **4. The description of Theodoridis et al. in the references includes what appear to be two separate citations.**
>
> Our understanding is that these are two distinct publications with some overlap between them. Theodoridis et al. (2025a) [A] is a paper describing in detail the dataset that was used in Theodoridis et al. (2025b) [B]. Although the two are connected, they appear to be two different papers.
>
> We added the following footnote in section 5.2.3 to make the overlap between the two papers clear:
>
> > This paper describes the dataset used in Theodoridis et al. (2025b), with substantial overlap between the two papers.

---

> ### Author Response · Authors · 2026-06-05
>
> ### Non-Critical
>
> **1. The language describing orientation encoding should be made more precise in the abstract, introduction, and conclusion...The revised text should clarify that implicit encoding of orientation is model-dependent and distance-limited, with Ovis2.5 being the strongest case.**
>
> We agree that the original wording was somewhat misleading, and we have revised the text accordingly in the following sections:
>
> - In the abstract, we originally stated that “...spatial visual concepts, such as the orientation of an object or agent, are only implicitly encoded by the spatial structure retained by the vision encoder.” We revised this to “...spatial visual concepts, such as the orientation of an object or agent, are only implicitly encoded by the spatial structure retained by the vision encoder, or are not linearly encoded at all,” to clarify that implicit encoding is not always guaranteed.
>
> - In the Introduction (end of the fifth paragraph), the original text stated “However, spatial concepts are implicitly represented through the spatial structure retained in the vision encoder’s representations,” which suggests that this holds universally, including for orientation. We revised this to “However, in some cases and especially for short distances, spatial concepts are implicitly represented through the spatial structure retained in the vision encoder’s representations,” to make clear that this does not apply to all spatial concepts and all settings.
>
> - In the conclusion, there was no reference to orientation being implicitly encoded, so no changes were necessary.
>
> We chose not to explicitly mention that Ovis2.5 represents the strongest case, as we consider such details are premature for the abstract and Introduction, and it is later discussed in section 5.
>
> **2. The activation steering experiments in Table 3 would benefit from a brief quantitative summary.**
>
> We agree that a quantitative summary is needed. To address this, we have added Table 6 in Appendix C, which reports activation steering success rates for all models and data categories on 50 randomly selected samples from the dataset.
>
> **3. The Appendix C analysis of cognitive failure causes is described by the authors themselves as "early exploration," yet it is presented at the same level of formality as the main results.**
>
> We do not consider our preliminary exploration of the causes of cognitive failure to be presented with the same level of formality as the main results, as it is discussed exclusively in the appendix. We believe that this sufficiently frames this as a preliminary investigation. Therefore, we have not made any changes to the paper in response to this comment.
>
> **4. The paper would benefit from a brief discussion of the potential influence of CARLA's visual statistics on the results.**
>
> We added the following text in the "Limitations and Future Work" section (Section 6):
>
> > Another limitation of this paper is that it is based only on synthetic data, as it is difficult to obtain identical real traffic scenes at scale, where the only difference is the targeted visual concept. CARLA produces images with simplified texture statistics, more uniform material properties, and a narrower distribution of object appearances compared to real-world scenes (Pasios & Nikolaidis, 2025). As a result, we cannot be certain that the visual concepts we find to be encoded in the models would also be encoded in the same way when using real images. The out-of-distribution evaluation, the results of which are presented in Table 4, partially addresses this concern for presence and count, but the same is not true for spatial-relationship and orientation. Therefore, another direction for future work would be to include real data for all data categories.

---

> ### Author Response · Authors · 2026-06-05
>
> **5. The rationale for selecting these three specific model families over other equally available sub-4B models (such as PaliGemma-3B, Phi-3.5-Vision, or MiniCPM-V) deserves a sentence or two of justification beyond performance ranking.**
>
> Besides their SOTA performance, Ovis2.5, InternVL3.5, and Qwen3-VL (added after the request of reviewer 6Xnc), were chosen because they exhibit key architectural differences (e.g. projector size, whether they split the input image into tiles, or if they use visual features from multiple layers of the vision encoder) that allow direct comparisons of how these design choices may influence the linear encoding of the visual concepts we study. Although we discussed these differences in the paper, we did not explicitly state that they motivated our model selection. We have therefore added the following text to the second paragraph of Section 4.1:
>
> > We selected Ovis2.5 (Lu et al., 2025), InternVL3.5 (Wang et al., 2025), and Qwen3-VL (Bai et al., 2025) because they are all well-known SOTA models and exhibit key architectural differences, as discussed below, which enable direct comparisons of how these differences influence the linear encoding of the studied visual concepts.
>
> Regarding VST, we already stated that it was chosen due to the additional spatial SFT and RL training it underwent, and we consider this sufficient justification for its inclusion.
>
> Finally, regarding the newly added DriveFusionQA model, its inclusion is justified in the paper as being used to check whether the patterns observed in general purpose VLMs still hold for models fine-tuned for autonomous driving tasks.
>
> **6. Vompa et al. (2025) should be discussed more specifically at the point in Section 5.3 where Figure 7 is introduced.**
>
> We made a small change in Section 5.3. Instead of comparing the accuracy of the models to the accuracy of linear probes trained on the concatenated activations of the visual tokens **and** the last token in the sequence, we now compare the accuracy of the models to linear probes trained **only** on the activations of the last token in the sequence. We believe that this comparison better supports the distinction between perception and cognitive failure, as both predictions, that is, the generative prediction of the model and the binary prediction of the linear probe, are based on exactly the same latent representation (that of the last token in the sequence after the post layernorm). Given this change, below is our answer regarding whether Vompa et al. (2025) should be discussed at this point:
>
> The alignment gap of Vompa et al. is between the generative performance of the model and the linear separability of the visual representations right after the vision encoder. They also provide some detailed results where we can see the comparison between the generative accuracy of the model and the linear separability of the visual representations at the last LLM layer. There we can see that indeed in many cases the models underperform even compared to the linear separability of their visual representations at the last layer. However, this differs from what we show in Figure 7 in two main points:
>
> - The accuracy of the probes in Figure 7 is based on the activations of the last token in the sequence, not on the visual tokens.
>
> - The accuracy of the probes in Figure 7 is based on the activations of the model after the final normalization layer, at which point, any information encoded in the activations of the final token will need to be linearly encoded if it is to be of any use for the model (as the generative process itself simply passes these activations through a linear layer, the language head).
>
> Because of these differences, we believe that, although related, the findings of Vompa et al. (2025) are not exactly the same and do not provide an explanation for the cognitive failure as presented here. Therefore, we believe that mentioning this paper in the "Related Work" section is sufficient.
>
> ### Additional Concerns Mentioned
>
> **VST (spatially tuned) models do not substantially outperform general-purpose models. This finding is implicit rather than explicitly foregrounded.**
>
> We realized that this is an important point that should be stated explicitly in the paper. We therefore added the following sentence at the end of the third paragraph in Section 5.1.6:
>
> > Finally, the two VST models and DriveFusionQA broadly exhibit the same patterns as the other models, indicating that fine-tuning and RL training on spatial or autonomous driving tasks do not materially change the intermediate visual concept encoding patterns.
>
>
> [A] https://arxiv.org/abs/2511.13397
>
> [B] https://ieeexplore.ieee.org/document/11230063

---

### Review · Reviewer_gAtL · 2026-05-21

**Summary Of Contributions:**

The paper studies how visual concepts are encoded and evolve across the various modules and layers of recent vision-language models (VLMs). In particular, it focuses on (synthetic) images related to autonomous driving tasks, and concepts (presence, counting, spatial relationship and orientation) relevant in such context. Counterfactual images are generated for each concept (e.g. image pairs with and without a given object to study the presence concept), and varying the distance of the object from the point of view. These are used to train linear probes which rely on the activations at different layers to classify input images along each concept. Analyzing the accuracy of the linear probe classifiers across layers and modules, the paper distinguishes cases of perceptual failure, where the visual information is not linearly encoded in the activation space (i.e., the linear probe fail), and cognitive failure, where the visual information is present but the model provides an incorrect answer. The quality of the obtained linear probes is further tested via activation steering and out-of-distribution evaluation.

Strengths
- Interpreting the interplay between modalities and the evolution of visual information in VLMs as well as understanding their failure modes are relevant problems.

- The experimental evaluation about autonomous driving tasks is relatively extensive, with several tasks and the ablation about object distance.

Weaknesses
-  The novelty, both in terms of technique and observations, seems limited. Inspecting how VLMs handle visual concepts with linear probes is a popular approach. Moreover, I think the various observations about perceptual vs cognitive failures are discussed in some form in prior works, as also mentioned in the paper. Also, the fact that smaller (more distant) objects make the tasks more difficult is not surprising. The fact that VLMs (and especially encoders) struggle with spatial relationship tasks is also known [A].

- The paper doesn't seem to provide new or actionable insights: for example, no method is proposed to counter any of the failure modes discussed.

- Some statements and evaluation methods are not convincing or well supported.
  - The paper states that counterfactual images are necessary (Sec. 3), but it's not clear why using different images with and without pedestrians (e.g. for the Presence-1 concept) wouldn't work.
  - The region pooling approach is quite arbitrary, and assumes that the information regarding e.g. spatial relationship is encoded in the same patches as the target objects in the image, which is unclear especially for activations far from the input layer.

[A] https://arxiv.org/abs/2506.03096

**Audience:**

Yes

**Audience Explanation:**

Interpreting the interplay between modalities and the evolution of visual information in VLMs as well as understanding their failure modes are relevant problems.

**Claims And Evidence:**

Yes

**Claims Explanation:**

The experimental evaluation is carried out, overall, in a systematic and consistent way, considering several models and visual concepts. This reveals the different failure modes of VLMs claimed by the paper.

**Requested Changes:**

As mentioned above (see weaknesses), I think the major weakness is the lack of novel (actionable) insights or methods. The paper should clarify what its novel contributions are, as at the moment it seems to only confirm previous observations on a specific task, with minor ablations (e.g. distance) with expected results. Moreover, some statements and design choices for the evaluation setup should be better motivated.

---

> ### Author Response · Authors · 2026-06-05
>
> We sincerely thank the reviewer for the thorough review. We address each point below.
>
> ---
>
> **1. The novelty, both in terms of technique and observations, seems limited.**
>
> We agree that using linear probes to study the intermediate activations of deep models is a well-established technique. However, we argue that, although the technique itself is not novel, our methodological contribution lies in the level of granularity with which linear probes are applied to VLMs and in the fine-grained targeting of specific visual concepts through carefully constructed counterfactual examples. More specifically, the following aspects differentiate our work from prior studies:
>
> 1. We perform a detailed layer-wise analysis of the intermediate activations across the entire model, including the vision encoder, which is omitted in most previous works.
>
> 2. Textual counterfactual inputs, where all tokens remain identical except for a single word, have been widely used in LLM interpretability research (e.g., [A]). However, to the best of our knowledge, no prior study has leveraged the equivalent concept of high-quality visual counterfactuals (i.e., identical images that differ only in a single thing) for VLM interpretability.
>
> 3. While training linear probes is a well-established practice, it is not clear how intermediate activations should be reduced in dimensionality in order to be able to train probes on them. In the paper, we provide a detailed discussion of our activation extraction methods and justify each choice based on the visual concepts being investigated. Furthermore, we introduce region pooling as an activation extraction method for better understanding how visual information relevant to spatial tasks is preserved and processed throughout the model.
>
> Regarding the observations, we believe that our study does present novel observations. We discuss this in greater detail in our response to point 5.
>
> **2. The various observations about perceptual vs cognitive failures are discussed in some form in prior works.**
>
> We agree that observations similar to ours have been discussed in prior work. To address this point, we made a small change in Section 5.3. Instead of comparing the accuracy of the models to the accuracy of linear probes trained on the concatenated activations of the visual tokens **and** the last token in the sequence, we now compare the accuracy of the models to linear probes trained **only** on the activations of the last token in the sequence. We believe that this comparison better supports the distinction between perceptual and cognitive failure, as both predictions, that is, the generative prediction of the model and the binary prediction of the linear probe, are based on exactly the same latent representation. This differs from the analyses presented in previous papers and provides more direct evidence of cognitive failure. Below, we explain why this is the case and why the evidence presented in the revised paper offers more direct support for cognitive failure than the studies reporting similar observations (discussed in Section 2.4).
>
> **Zhang et al. (2024)** and **Vompa et al. (2025)** use only the intermediate activations corresponding to the visual tokens and not the activations of the last token in the sequence. Consequently, it is possible that a concept is encoded in the visual-token activations but never becomes encoded in the representation of the last token, which is the representation that directly influences the model's generated answer. **Fu et al. (2025b)** do not describe their methodology in sufficient detail to determine exactly which LLM activations were used, but the most conservative interpretation is that they also relied exclusively on visual-token activations. In contrast, we make a stronger claim by showing that visual information can remain linearly encoded in the activations of the last token itself, particularly after the final layer normalization, where only a linear projection (the language head) separates the representation from the generated answer. Despite this, the model may still fail to answer correctly. This observation more precisely localizes the cognitive failure as a failure to map encoded visual information to the appropriate linguistic semantics.

---

> ### Author Response · Authors · 2026-06-05
>
> **3. The fact that smaller (more distant) objects make the tasks more difficult is not surprising.**
>
> We definitely agree that distant objects being less linearly encoded is not a surprising result, and we did not intend to present it as such. We explicitly acknowledged this in Section 5.1.5 (5.1.6 in the revised paper):
>
> > ...it is unreasonable to expect a vision encoder to represent a pedestrian equally well at 5 meters and 50 meters, particularly when it has no prior knowledge of the question these features will be used to answer. Therefore, part of the failure at longer distances can also be attributed to the LLM, which does not sufficiently compensate for degraded visual representations.
>
> However, we believe that analysing how distance affects the encoding of different visual concepts within the model's intermediate layers is valuable, particularly for automated driving tasks where the distance of other objects and agents is critical. To the best of our knowledge, this is the first study to examine this phenomenon at such a level of detail.
>
> **4. The fact that VLMs (and especially encoders) struggle with spatial relationship tasks is also known.**
>
> We agree that this is considered a well-established finding. However, we do not present it as a contribution of our paper. Rather, our contribution lies in analyzing how visual information related to spatial relationships is encoded throughout the architecture and investigating whether this information is encoded explicitly or implicitly by repeating the experiments with both average pooling and region pooling. To the best of our knowledge, this aspect has not been studied previously.
>
> **5. The paper doesn't seem to provide new or actionable insights. No method is proposed to counter any of the failure modes discussed.**
>
> We believe that the paper provides several new insights, which we summarize below:
>
> 1. Our results show that LLMs can explicitly and linearly encode spatial information even when that information is not explicitly and linearly encoded by the vision encoder. This can be observed, for example, in the increase in probe accuracy across the middle and later LLM layers for both spatial tasks in all evaluated VLMs. We hypothesize that this occurs because the LLM leverages spatial information that remains implicitly encoded within the vision encoder (Figure 5).
>
> 2. On the other hand, our results show that LLMs are not capable of explicitly and linearly encoding orientation information, even when this information appears to be implicitly encoded by the vision encoder in some cases (e.g., Orientation-2 at short distances). We now explain this more clearly in the revised paper in the last paragraph of section 5.1.4.
>
> 3. Our results show that the models encode not only the target concept as it appears in specific categories, but also the underlying visual concept itself to some degree, as demonstrated by the results in Figure 6. To the best of our knowledge, this finding has not been previously reported in the literature.
>
> 4. Our results show that VST-SFT and VST-RL exhibit similar encoding patterns to the rest of the models, without any advantage on the spatial relationship and orientation tasks. That shows that finetuning on spatial data doesn't necessarily change how spatial visual concepts are encoded within the model (i.e., whether they are linearly encoded or not). This insight wasn't explicitly mentioned, but we now mention it explicitly in the revised paper (section 5.1.6), along with the same conclusion for the newly added (after the request of reviewer 6Xnc) DriveFusionQA, a VLM fine-tuned for autonomous driving applications.
>
> Additionally, the revised version of the paper provides stronger evidence for cognitive failure, as discussed in point 2.
>
> Regarding potential methods to address any of the failure modes discussed, we acknowledge that this is a limitation of the paper. At the same time, we consider the development of mitigation strategies to be beyond the scope of the present study. While methods for addressing some of these limitations would certainly be valuable, our focus in this work is on identifying and analysing the limitations themselves. We therefore leave the investigation of mitigation strategies to future work. We now explicitly state this in Section 6 (Limitations and Future Work) of the revised paper.

---

> ### Author Response · Authors · 2026-06-05
>
> **6. It's not clear why using different images with and without pedestrians (e.g., for the Presence-1 concept) wouldn't work.**
>
> We agree that using real images that are not identical in every aspect apart from the target concept might have been possible; however, it would have a few drawbacks compared to using high-quality counterfactuals:
>
> - Most of the traffic scenes datasets don't include annotations for very distant (but visible) objects. That would mean that we would need to manually verify all samples in order to achieve the same level of ground truth annotations quality that the simulated data provide.
>
> - For Spatial-1, autonomous driving datasets don't provide annotations for the blinker state, and for Spatial-2, pedestrians can be in any place in the scene, which would have made replicating these categories impossible.
>
> - The region-pooling method would not be feasible with real-world data, as the location of a pedestrian, bicycle, or truck would vary across images even at the same distance. Consequently, a different splitting point would be required for each image.
>
> For these reasons, we believe that simulated data, which enable perfectly controlled counterfactuals, are the most appropriate choice for the analysis presented in this paper. We have now revised the first paragraph of Section 3 to better motivate this design choice, as discussed in greater detail in the requested changes below.
>
> **7. The region pooling approach is quite arbitrary, and assumes that the information regarding, e.g., spatial relationship is encoded in the same patches as the target objects in the image.**
>
> The idea behind region pooling is not that the patches themselves encode spatial relationships (or orientation). Rather, the assumption is that they retain local information (which has been shown to be true [B]). For example, if the patch at row 7 and column 13 encodes a face and the patch at row 7, column 14 encodes the back of a head, this information can later be combined by the LLM layers to infer the pedestrian's orientation (in this example, facing left).
>
> To clarify why we believe this is a useful experiment for understanding what went wrong in the model, consider Ovis2.5 on Orientation-1 at 5 meters. If we based our conclusions solely on the average-pooling results (Figure 4), we would infer that the vision encoder is primarily responsible for the model's failure to encode pedestrian orientation. In other words, orientation is not linearly encoded in the vision encoder's representations, leaving the LLM component with no relevant information to exploit.
>
> However, Figure 5 suggests a different interpretation. The vision encoder appears to retain sufficient spatial information for pedestrian orientation to be inferred, even if that information is not explicitly encoded. Therefore, we can argue that the vision encoder does not explicitly encode pedestrian orientation, which is not particularly surprising given that it has no knowledge of the downstream question, but it does preserve enough information for the LLM component, once conditioned on the task, to infer it.
>
> Taken together, these results suggest that the failure of Ovis2.5 to encode pedestrian orientation at 5 meters is primarily due to the LLM's inability to effectively combine the available visual information, rather than to the complete absence of that information in the vision encoder's representations.

---

> ### Author Response · Authors · 2026-06-05
>
> ### Requested changes:
>
> **1. The paper should clarify what its novel contributions are.**
>
> We have revised the paper to make our methodological contributions (as discussed in Point 1 above) and the new insights (as discussed in Point 5 above) clearer.
>
> **Methodological contributions**
>
> 1. We now state more explicitly in the Introduction that we analyse the flow of visual information **throughout the entire architecture**.
> 2. We added a contribution statement in the Introduction that explicitly highlights the use of perfect visual counterfactuals.
>
> **New insights**
>
> 1. We revised the paper to more clearly present the finding that LLMs are not capable of explicitly and linearly encoding orientation information, even when this information appears to be implicitly encoded within the vision encoder (Section 5.1.4, last paragraph).
> 2. We also more clearly present the finding that fine-tuning on spatial or autonomous-driving data does not materially alter the model's encoding patterns (Section 5.1.6, third paragraph).
>
> We believe that the remaining methodological contributions and new insights discussed in Points 1 and 5 above, respectively, are already presented sufficiently clearly in the paper. Therefore, we did not make any further changes in response to this comment.
>
>
> **2. Some statements and design choices for the evaluation setup should be better motivated.**
>
> Regarding design choices, we consider the following to be the main design choices of our paper:
>
> - **The use of perfect counterfactuals instead of real images.** We agree that this design choice could be better motivated. Accordingly, we revised the first paragraph of Section 3. Specifically, we now present this choice as analogous to the use of textual counterfactual inputs in LLM interpretability research, where carefully controlled counterfactuals differ only in the target concept of interest.
>
> - **The study of four basic visual concepts: presence, count, spatial relationship, and orientation.** We believe that this choice is already sufficiently motivated in Section 3.1:
>
>     >  A typical traffic scene contains multiple objects or agents, which relate to presence and count, arranged within a three-dimensional environment, which relates to spatial relationships, and exhibiting various orientations. We therefore argue that successfully encoding these visual concepts is necessary, although not sufficient, for a proper understanding of a traffic scene.
>
> - **The activation extraction methodology.** We believe that this design choice is also sufficiently motivated in Section 4.3 of the paper.
>
> Regarding the comment about statements that require stronger motivation, it is not clear to us which specific statements the reviewer is referring to. Consequently, we did not make any changes in response to this part of the comment.
>
> [A] https://arxiv.org/abs/2211.00593
>
> [B] https://arxiv.org/abs/2410.07149

---

> > ### Comment · Reviewer_gAtL · 2026-06-16
> >
> > I thank the authors for the detailed response. I would agree that there are some small methodological differences with prior works, but I think these don't provide any substantially different insights, but rather confirm previous observations (which is largely acknowledged in the rebuttal). Moreover, it's not clear how or if the findings of the paper could be used to improve the limitations of the VLMs. Therefore, while the experimental evaluation is quite extensive, I think the contributions of the paper are limited.

---

> > > ### Author Response · Authors · 2026-06-18
> > >
> > > We thank the reviewer for the follow-up. We agree that some of the insights are related to those of previous works, such as the observations about perceptual and cognitive failure, although we provide stronger evidence for this distinction. However, we would like to clarify that our rebuttal was not intended to acknowledge that the paper’s observations are simply confirmatory of previous work. More specifically, under point 5 in our previous reply, we listed four observations which, to the best of our knowledge, have not been previously reported in the literature. We believe these findings are particularly relevant in the automated driving setting, where failures in the encoding of basic concepts such as orientation can directly affect the interpretation of traffic scenes. Understanding how the interplay between the modules of VLMs fails to encode such concepts is therefore valuable. If a further revision is requested, we would be happy to try to make those observations clearer in the paper.

---

### Review · Reviewer_6Xnc · 2026-05-22

**Summary Of Contributions:**

The paper analyzes where small vision‑language models lose visual information for driving. Using CARLA counterfactuals and linear probes across layers, it shows presence/count are often linearly encoded, but spatial/orientation cues are weak and degrade with distance. Comparing probes to model outputs reveals two failure modes: perceptual and cognitive, highlighting bottlenecks and guidance for improving lightweight VLMs.

Strengths:
1. The paper is well written and easy to follow.
2. The research questions are clear, targeted, and highly relevant to autonomous driving.
3. Several experimental findings are particularly interesting.

Weaknesses:
1. The paper primarily applies existing, standard techniques to the autonomous driving setting, offering limited methodological contribution.
2. Most conclusions are already well known. For example, it is widely recognized that CLIP‑style pretraining struggles with fine‑grained details, spatial relationships, and orientation, and that aligning visual features with language representations is a major bottleneck for VLMs. The paper largely restates these points from an autonomous‑driving perspective.
3. The models evaluated are all general‑purpose and not specifically trained or fine‑tuned for autonomous driving, making it unclear whether the conclusions transfer to AD applications.
4. The probing dataset is relatively small, and the set of studied concepts lacks diversity.
5. The paper mainly catalogs a few VLM failure cases in AD but offers little analysis of their root causes or concrete mitigation strategies, limiting its practical value.

**Audience:**

Yes

**Audience Explanation:**

The paper investigates key challenges in applying VLMs to autonomous driving, with direct implications for model safety and trustworthiness. Even if methodological novelty is limited, the controlled counterfactual setup and component‑wise diagnostics in a safety‑critical domain make the results useful and informative.

**Claims And Evidence:**

No

**Claims Explanation:**

The paper offers comprehensive experiments and thoughtful analysis, but the small dataset and limited statistical treatment reduce confidence in the results.

**Requested Changes:**

1. It would be better to also experiment on some VLMs specifically trained or finetuned for autonomous driving applications.
2. Since all three models studied in this model use the Qwen family LLMs, it would be nice also to include the results of Qwen-VL.
3. For the experiments of activation steering, it would be good if the authors could provide some quantitative evaluation, such as success rate, in addition to the current examples.

---

> ### Author Response · Authors · 2026-06-05
>
> We sincerely thank the reviewer for the thorough review. We address each point below.
>
> ---
>
> ### Weaknesses
>
> **1. The paper primarily applies existing, standard techniques to the autonomous driving setting, offering limited methodological contribution.**
>
> We agree that using linear probes to study the intermediate activations of deep models is a well-established technique. However, we argue that, although the technique itself is not novel, our methodological contribution lies in the level of granularity with which linear probes are applied to VLMs and in the fine-grained targeting of specific visual concepts through carefully constructed counterfactual examples. More specifically, the following aspects differentiate our work from prior studies:
>
> 1. We perform a detailed layer-wise analysis of the intermediate activations across the entire model, including the vision encoder, which is omitted in most previous works.
>
> 2. Textual counterfactual inputs, where all tokens remain identical except for a single word, have been widely used in LLM interpretability research (e.g., [A]). However, to the best of our knowledge, no prior study has leveraged the equivalent concept of high-quality visual counterfactuals (i.e., identical images that differ only in a single thing) for VLM interpretability.
>
> 3. While training linear probes is a well-established practice, it is not clear how intermediate activations should be reduced in dimensionality in order to be able to train probes on them. In the paper, we provide a detailed discussion of our activation extraction methods and justify each choice based on the visual concepts being investigated. Furthermore, we introduce region pooling as an activation extraction method for better understanding how visual information relevant to spatial tasks is preserved and processed throughout the model.
>
> **2. Most conclusions are already well known.**
>
> We believe that the paper provides several new insights, which we summarize below:
>
> 1. Our results show that LLMs can explicitly and linearly encode spatial information even when that information is not explicitly and linearly encoded by the vision encoder. This can be observed, for example, in the increase in probe accuracy across the middle and later LLM layers for both spatial tasks in all evaluated VLMs. We hypothesize that this occurs because the LLM leverages spatial information that remains implicitly encoded within the vision encoder (Figure 5).
>
> 2. On the other hand, our results show that LLMs are not capable of explicitly and linearly encoding orientation information, even when this information appears to be implicitly encoded by the vision encoder in some cases (e.g., Orientation-2 at short distances). We now explain this more clearly in the revised paper in the last paragraph of section 5.1.4.
>
> 3. Our results show that the models encode not only the target concept as it appears in specific categories, but also the underlying visual concept itself, as demonstrated by the results in Figure 6. To the best of our knowledge, this finding has not been previously reported in the literature.
>
> 4. Our results show that VST-SFT and VST-RL follow encoding patterns comparable to the other models and do not demonstrate any performance advantage on the spatial relationship and orientation tasks. That shows that finetuning on spatial data doesn't necessarily change how spatial visual concepts are encoded within the model (i.e., if they are linearly encoded or not). This insight wasn't explicitly mentioned, but we now do mention it explicitly in the revised paper (section 5.1.6), along with the same conclusion for the newly added DriveFusionQA (requested change 1 below).
>
> Additionally, the revised version of the paper provides stronger evidence for cognitive failure, as discussed in our response to reviewer gAtL (point 2).
>
> **3. The models evaluated are all general‑purpose and not specifically trained or fine‑tuned for autonomous driving.**
>
> We agree that the inclusion of at least one fine-tuned model for autonomous driving makes sense. We added the DriveFusionQA [B] model in our evaluations as discussed in the requested changes below.

---

> ### Author Response · Authors · 2026-06-05
>
> **4. The probing dataset is relatively small, and the set of studied concepts lacks diversity. | the small dataset and limited statistical treatment reduce confidence in the results.**
>
> First, regarding the size of the probing dataset, it contains approximately 55,000 samples in total (500 samples per class per distance for each category, excluding the negative samples for the Presence and Count concepts, which are not distributed across six distance bins). We therefore believe that the dataset is sufficiently large for a probing study.
>
> However, most of these samples are used for training and validation, while only a relatively small subset is reserved for testing the probes, from which the results reported in the paper are obtained. We therefore agree with the reviewer that additional statistical analysis can help increase confidence in the findings. In the original version of the paper, the statistical treatment was limited to repeating the linear probe training, validation, and testing procedure ten times and reporting the mean test accuracy across the ten runs, together with the corresponding standard deviation. Specifically, we stated:
>
> > To reduce stochastic fluctuations in accuracy across layers, we repeat this process ten times and report the average test accuracy of the best probe from each run, along with the standard deviation across the ten runs.
>
> ...
>
> > The standard deviation of probe accuracies across the ten runs for a fixed model, data category, layer, and distance is generally low, ranging from 0.00 to 0.09, with an average value of 0.02.
>
> In the revised version of the paper, we complement this analysis with an additional statistical assessment. Specifically, for every probe chance-corrected accuracy reported in Figures 4 and 5, we compute the lower bound of the 95% Wilson score confidence interval (i.e., we first compute the lower bound for the observed accuracy and then derive the corresponding chance-corrected accuracy from that lower bound) and include the resulting figures in the appendix. These results show that the key trends and conclusions discussed in the main text remain unchanged even under this conservative estimate of performance. Across all reported results, the average difference between the observed accuracy and the lower bound of its 95% Wilson confidence interval is 7.93 pp (percentage points), with a standard deviation of 2.13 pp. More details are reported in Appendix B of the revised paper. We hope that this additional analysis addresses concerns regarding the statistical reliability of the reported findings.
>
> Regarding the diversity of the visual concepts studied, we already acknowledged in the paper that the number of concepts considered is limited. However, our objective was to focus on a set of fundamental visual concepts that are directly relevant to understanding traffic scenes. Given this focus, we believe that the range of additional concepts that could be included while remaining sufficiently basic is relatively limited. Furthermore, concentrating on a smaller set of core concepts enabled us to perform a much more detailed analysis of each one across model components, layers, and object distances. We therefore view this as a deliberate trade-off between breadth and depth.
>
> **5. The paper mainly catalogs a few VLM failure cases in AD but offers little analysis of their root causes or concrete mitigation strategies, limiting its practical value.**
>
> Regarding root causes, we include a preliminary investigation into the reasons for cognitive failure in Appendix D, identifying surface form competition [C] as a potential reason.
>
> Regarding mitigation strategies, we acknowledge that this is a limitation of the paper. At the same time, we consider the development of mitigation strategies to be beyond the scope of the present study. While methods for addressing some of these limitations would certainly be valuable, our focus in this work is on identifying and analysing the limitations themselves. We therefore leave the investigation of mitigation strategies to future work. We now explicitly state this in Section 6 (Limitations and Future Work) of the revised paper.

---

> ### Author Response · Authors · 2026-06-05
>
> ### Requested changes
>
> **1. It would be better to also experiment on some VLMs specifically trained or finetuned for autonomous driving applications.**
>
> We agree that including at least one autonomous driving–specific fine-tuned model is valuable. Accordingly, we added DriveFusionQA [B] to our evaluation in the revised paper. As discussed in the manuscript, the results suggest that fine-tuning on autonomous driving data alone does not alter how basic visual concepts are encoded within the model.
>
> **2. Since all three models studied in this paper use the Qwen family LLMs, it would be nice also to include the results of Qwen-VL.**
>
> We agree that including a Qwen-VL model is appropriate, given that some of the evaluated models are also based on the Qwen architecture. Accordingly, we added Qwen3-VL-2B-Instruct [D] to our evaluations in the revised paper. As shown in the results, the encoding of visual concepts in this model exhibits patterns that are highly consistent with those observed across the other evaluated models.
>
> **3. For the experiments of activation steering, it would be good if the authors could provide some quantitative evaluation**
>
> We agree that a quantitative evaluation is needed. To address this, we have added Table 6 in Appendix C, which reports activation steering success rates for all models and data categories on 50 randomly selected samples from the dataset.
>
> [A] https://arxiv.org/abs/2211.00593
>
> [B] https://huggingface.co/DriveFusion/DriveFusionQA
>
> [C] https://arxiv.org/abs/2104.08315
>
> [D] https://huggingface.co/Qwen/Qwen3-VL-2B-Instruct

---

### Author Response · Authors · 2026-06-05

We sincerely thank the reviewers for their evaluation of the manuscript. We have addressed their concerns in individual responses under each review. We have also uploaded a revised version of the paper, together with a diff document (under supplementary material) highlighting the changes between the original and revised manuscripts. Below, we discuss a minor methodological change introduced in the revised paper. This modification does not affect the observed patterns or the main conclusions; however, we believe it is important to explain the motivation behind it.

---

When extracting activations from Qwen3-VL, following the request of Reviewer 6Xnc, we observed that the variance of activation dimensions across samples can differ substantially between layers. This variation can potentially affect the conditioning of the probe optimization problem, making comparisons across layers less reliable. To mitigate this issue, we standardized the activations before training the probes across all models and data categories. Specifically, we compute the mean and standard deviation on the training set and use them to standardize the activations during training, validation, and testing. As a result, each activation dimension has zero mean and unit variance across the training samples, improving the conditioning of the probe-training problem and enabling fairer comparisons across layers.

Importantly, this change does not alter the underlying linear decodability question. Standardization is an affine transformation, and because our linear probes include a bias term, a concept is linearly decodable from the raw activation space if and only if it is linearly decodable from the standardized activation space. Thus, the representational question remains fundamentally the same; the change only improves the numerical conditioning and comparability of probe training across layers.

Finally, since the results and observed patterns remain largely unchanged, we do not consider this a major revision. We include this clarification to explain the motivation for the change and to make the revised methodology more transparent.

---

### Decision · Action_Editor_MeGA · 2026-06-24

**Recommendation:** Accept with minor revision

**Additional Comments:**

In the revision, please address the comments from Reviewer 8qgi by providing a more thorough discussion of DriveFusionQA’s Count-1 and Count-2 results, and by clarifying how Qwen3-VL’s architecture affects the interpretation of the heatmaps. In addition, please consider adding a data-availability statement clarifying whether the dataset will be released, and if so, how it will be made available.

**Audience:**

Yes

**Audience Explanation:**

The paper received mixed reviews on this dimension. Specifically, Reviewers 8qgi and 6Xnc support the paper that it contributes by establishing a well-calibrated dataset and conducting experiments to demonstrate model capabilities in automated-driving tasks, while Reviewer gAtL raises concerns that many of the observations and insights may have appeared in previous work. In addition, although generally supportive, Reviewer 6Xnc also notes concerns regarding the paper’s novelty and some of the methodologies have already been explored. The AE checks the paper and the discussion and finds that, while the paper’s experimental method, such as linear probing, has been well established in previous work, the dataset and experiments support an application-domain contribution in automated driving. The paper brings existing tools into the automated-driving setting: it constructs counterfactual traffic-scene data for driving-relevant visual concepts, probes both lightweight general-purpose VLMs and AD-specific models. As such, the AE believes that part of the TMLR audience would still benefit from this work.

**Claims And Evidence:**

Yes

**Claims Explanation:**

In the initial review, Reviewer 6Xnc raised concerns regarding the relatively small dataset and limited statistical treatment. During the discussion, the authors provided a detailed clarification of the dataset size and added an additional statistical assessment in the revision. In addition, following reviewers’ suggestions, the authors added experiments on additional models, including an AD-specific model. After the discussion phase, all reviewers agreed that the submission is properly supported by evidence.

---

> ### Author Response · Authors · 2026-06-29
>
> Dear Editor,
>
> Thank you for the feedback. We would appreciate some clarification regarding the requested changes. From what we can see, there are no additional comments from Reviewer 8qgi discussing DriveFusionQA or Qwen3-VL. We were wondering whether these comments may not be visible to us. Could you please clarify this so that we can address the reviewer's comments appropriately?